# Complex transcriptional regulation and independent evolution of fungal-like traits in a relative of animals

**Alex de Mendoza[1]\*[†], Hiroshi Suga[1,2], Jon Permanyer[3,4], Manuel Irimia[3,4], Iñaki Ruiz-Trillo[1,5]\***

[1]Institut de Biologia Evolutiva, Universitat Pompeu Fabra, Barcelona, Spain; [2]Prefectural University of Hiroshima, Shobara, Japan; [3]EMBL-CRG Systems Biology Unit, Centre for Genomic Regulation, Barcelona, Spain; [4]Universitat Pompeu Fabra, Barcelona, Spain; [5]Institució Catalana de Recerca i Estudis Avançats, Barcelona, Spain

**\*For correspondence:**
alexmendozasoler@gmail.com
(AdM); inaki.ruiz@ibe.upf-csic.es
(IRT)

**Present address:** [†]ARC CoE
Plant Energy Biology, University
of Western Australia, Crawley,
Australia

**Competing interests:** The
authors declare that no
competing interests exist.

**Reviewing editor:** Alejandro
Sánchez Alvarado, Stowers
Institute for Medical Research,
United States

**Abstract** Cell-type specification through differential genome regulation is a hallmark of complex multicellularity. However, it remains unclear how this process evolved during the transition from unicellular to multicellular organisms. To address this question, we investigated transcriptional dynamics in the ichthyosporean *Creolimax fragrantissima*, a relative of animals that undergoes coenocytic development. We find that *Creolimax* utilizes dynamic regulation of alternative splicing, long inter-genic non-coding RNAs and co-regulated gene modules associated with animal multicellularity in a cell-type specific manner. Moreover, our study suggests that the different cell types of the three closest animal relatives (ichthyosporeans, filastereans and choanoflagellates) are the product of lineage-specific innovations. Additionally, a proteomic survey of the secretome reveals adaptations to a fungal-like lifestyle. In summary, the diversity of cell types among protistan relatives of animals and their complex genome regulation demonstrates that the last unicellular ancestor of animals was already capable of elaborate specification of cell types.

## Introduction

The process by which multicellular animals develop from a unicellular zygote is believed to mirror the first evolutionary steps that led to the origins of animal multicellularity from a unicellular species (*King, 2004*). As the development of complex multicellularity is dependent upon differential genome regulation, the evolutionary onset of animal multicellularity must likewise have involved the appearance of differential genome regulatory capacities leading to distinct cell types. Many of the genes involved in the control of animal development and cell type identity, including signaling pathways and transcription factors (TFs), pre-date animal origins (*Sebe-Pedros et al., 2011*; *Sebé-Pedrós et al., 2012*; *Suga et al., 2012*; *Richter and King, 2013*). As these genes are known to be present in the genomes of the protistan relatives of animals, the unicellular holozoans (*King et al., 2008*; *Suga et al., 2013*; *Fairclough et al., 2013*), the evolution of complex multicellularity, with cell type-specific transcriptional programs, must have involved changes in gene regulation. Therefore, a key step in understanding the evolution of multicellularity will be to infer the regulatory complexity of the last common ancestor of all living animals.

To address ancestral regulatory complexity, it will be necessary to elucidate the molecular control of cell differentiation through development in the unicellular relatives of animals. Three distinct developmental modes that lead to transient simple multicellular forms have been described in the protistan relatives of animals (*Figure 1*) (*Sebé-Pedrós et al., 2013*; *Suga and Ruiz-Trillo, 2013*;

**eLife digest** All living animals are descended from a single-celled ancestor, and understanding how these ancestors became the first multicellular animals remains a major challenge in the field of evolutionary biology. An early breakthrough towards this goal was the realization that, even though they're mostly single-celled organisms, the closest living relatives of animals share most of the basic gene toolkit that animals use to support their multicellular lifestyles.

This shared toolkit also includes the genes that allow each specialized cell type in an animal (for example, a skin cell or liver cell) to express the subset of genes that it needs to fulfil its specific role. Discovering how the single-celled relatives of animals regulate these and other "multicellularity-related" genes during their life cycles is the next crucial step towards understanding how animals became multicellular.

*Creolimax fragrantissima* is a single-celled relative of animals. One stage in this organism's life cycle involves its nucleus (which contains its genetic material) replicating multiple times without the cell itself dividing. After this stage of development, new cells are formed, each receiving with a single nucleus, and released to live freely in the environment. Characterizing how *C. fragrantissima* regulates which genes are expressed during these two very different stages of development could shed new light on how multicellular animals evolved to regulate their genes in specific cell types. However, little is known about these processes in *C. fragrantissima*.

Now, de Mendoza et al. have both sequenced *C. fragrantissima*'s genome and analysed which genes are expressed during the stages of its life cycle. This analysis reveals that this organism regulates its gene expression in several ways that are more commonly associated with gene regulation in multicellular animals. Furthermore, when compared to two other living relatives of animals that have brief multicellular stages in their life cycles, de Mendoza et al. found that the three organisms expressed similar genes during these similar life cycle stages.

Furthermore, like fungi, *C. fragrantissima* digests its food externally and then absorbs the nutrients. Using a range of techniques, de Mendoza et al. identified the proteins involved in these processes and discovered that many had evolved independently from their counterparts in fungi. Furthermore, in some cases, the genes for these proteins had actually been acquired from bacteria via a process called lateral gene transfer.

Together these findings suggest that it was likely that the last single-celled ancestor of multicellular animals already had the biological ability to create different cell types. Understanding if the cell types found in single-celled species resemble cell types from simple animals, such as sponges and comb jellies, at a molecular level is the next step towards determining what the ancestor of living animals looked like.

Dayel et al., 2011). Colonial clonal development has been shown to involve differential regulation of a few multicellularity-related genes in the choanoflagellate *Salpingoeca rosetta* (*Fairclough et al., 2013*). On the other hand, the filasterean amoeba *Capsaspora owczarzaki* exhibits up-regulation of adhesion-related genes in its aggregative stage (*Sebé-Pedrós et al., 2013*). To date, however, there has been no molecular characterization of coenocytic development, a third and completely distinct mode of development observed in the ichthyosporeans. Ichthyosporeans are the earliest branching holozoan lineage (*Torruella et al., 2012*; *Paps et al., 2013*; *Torruella et al., 2015*), and coenocytic development is a shared feature within the group (*Glockling et al., 2013*; *Mendoza et al., 2002*). This developmental mode comprises a growth stage in which nuclei divide synchronously within a common cytoplasm before undergoing cellularization, followed by release of motile ameboid zoo-spores (*Suga and Ruiz-Trillo, 2013*; *Marshall et al., 2008*). These stages have distinct physiological and structural characteristics, the amoeboid stage is mono-nucleated, non-diving, and motile (*Figure 1B*), while the multinucleate stage has a cell wall, a big central vacuole, and does not move (*Figure 1C*). Despite being quite distinct from canonic animal development, coenocytic development and multinucleate cell types are found in some animal lineages, such as in Drosophila syncytial blastoderm (*Suga and Ruiz-Trillo, 2013*). Thus, in order to infer ancestral regulatory complexity by comparison of premetazoan developmental modes, it will be essential to obtain a detailed molecular characterization of ichthyosporean coenocytic development.

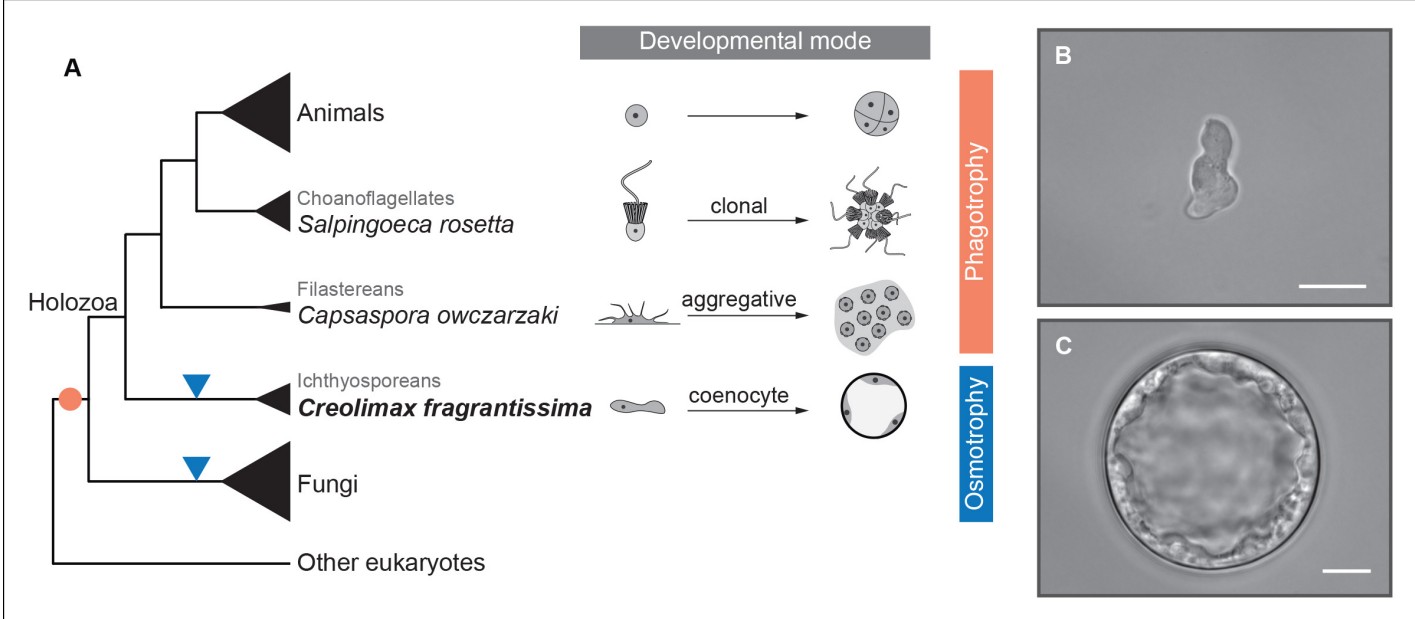

**Figure 1.** Evolution of developmental and feeding modes across holozoans. (**A**) The cladogram represents known phylogenetic relationships among holozoans (*Torruella et al., 2012*; *Torruella et al., 2015*). Each lineage is represented by the species proposed as a model system with a schema of its developmental mode on the right. The evolution of specialized osmotrophy is shown as a blue triangle in the cladogram, while the putative ancestral phagotrophic feeding mode of opisthokonta is shown as an orange circle (*Cavalier-Smith, 2012*). Divergence times of the lineages shown in this figure range between 700 Mya (considered the latest estimates of animal origins) and 1200 Mya (earliest estimates of Opisthokont origins) (Sharpe* et al., 2015). Micrographs depicting the (**B**) amoeboid stage and (**C**) multinucleate stage of *Creolimax fragrantissima* are shown. Scale bars = 10 μm. Choanoflagellate adapted from Mark Dayel (CC BY-SA 3.0) www.dayel.com/blog/2010/10/07/ choanoflagellate-illustration.

The following figure supplement is available for figure 1:

**Figure supplement 1.** Creolimax synchronized stages.

Here we describe the transcriptome dynamics of *Creolimax fragrantissima*, which has been proposed as a model system for ichthyosporeans (*Suga and Ruiz-Trillo, 2013*; *Marshall et al., 2008*). We show that *Creolimax* employs complex gene regulation including alternative splicing and the use of long intergenic non-coding RNAs (lincRNA). Through analysis of the *Creolimax* secretome in silico and by proteomics, we also provide evidence of secondary adaptation to a specialized osmotrophic feeding mode through lateral gene transfer (LGT) and gene duplication. Taken together, our results suggest that the last common unicellular ancestor of animals was already capable of implementing elaborate, cell type-specific differentiation programs.

## Results

### Assembly and annotation of a reference genome

As a reference genome for mapping RNA sequencing (RNA-seq) data, we assembled the 45 Mb draft genome of *Creolimax* from a combination of 454 reads and mate-paired end Illumina reads corresponding to a 75× coverage. The draft assembly comprises 82 scaffolds, with an N50 of 1.5 Mb (see Materials and methods). The genome is twice as large as that of *Capsaspora* but in line with sequenced choanoflagellate genomes. We annotated 8695 genes, 92% of which are supported by transcriptional evidence. A diverse array of annotation pipelines rendered functional information on 78% of the predicted protein coding genes (see Material and methods). Among those genes, many belong to gene families involved in multicellularity and development in animals, such as TFs and signaling pathways, as previously reported in targeted gene family analysis (*Suga et al., 2014*; *de Mendoza et al., 2013*; *de Mendoza et al., 2014*).

## Transcriptional dynamics reveal differences between *Creolimax* multinucleate stage and animal development

We isolated two different stages of the *Creolimax* lifecycle by taking advantage of the cell size difference between the amoeboid stage and the multinucleate growth stages (*Marshall et al., 2008*). After filtering with a 5 µm mesh, we obtained a highly enriched culture of amoebae. The amoebae encysted and grew for at least 24 hr (*Figure 1—figure supplement 1*), and then multinucleate coenocytic cysts matured asynchronously, releasing new amoeboid zoospores with different cell sizes to form a heterogeneous culture. Given the drastic morphological difference between the non-motile mitotic multinucleate stage pre-dominant in the 24 hr culture and the motile non-dividing amoeba isolated after filtration, we decided to investigate their transcriptomic differences through RNA-seq (see Material and methods).

The amoeboid and the multinucleate stage showed distinct transcriptional profiles, consistent among replicates (*Figure 2A*), from which we identified 956 genes as significantly differentially expressed. Functional enrichment analyses of Gene Ontologies (GOs) and PFAM domains (p<0.01, Fisher's exact test) revealed that the multinucleate stage shows up-regulation of genes associated with cell growth, including ribosome, translation, DNA replication, amino acid and RNA metabolism activities (*Figure 2*, *Figure 2—figure supplement 1*). Conversely, in the amoeboid stage, we found enrichment for signaling activities (GTPases and kinases) and an up-regulation of the actin cytoskeleton, most likely involved in the motile behavior of the amoeba (see raw GOs in *Figure 2—source data 1*). Strikingly, single-celled amoebas also showed up-regulation of extracellular matrix (ECM) adhesion, including the up-regulation of the integrin pathway components.

Besides signaling and adhesion activities, regulation through TFs is also considered to be a defining characteristic of animal development (*de Mendoza et al., 2013*; *Levine and Tjian, 2003*). To address whether that is the case for the multinucleate stage in *Creolimax*, we compared the stage-specific expression level of all the TFs encoded in the genome. Contrary to our expectation, the comparison revealed that TF expression is higher in the amoeboid stage (p<0.05, Wilcoxon signed rank test, *Figure 2—figure supplement 2A*). Among the 127 putative TFs found in the genome, only 5 are significantly up-regulated in the multinucleate stage. These five genes have DNA binding domains with ambiguous TF activity and include genes containing Myb_DNA-binding and cold-shock domains (*Figure 2—figure supplement 2B*). In the amoeboid stage, we found significant up-regulation of 11 TFs, including T-box, Runx and hemophagocytic lymphohistiocytosis TFs, and TF classes that have important roles in animal development (*Sebe-Pedros et al., 2011*). Altogether, these data indicate that, unlike in animal development, the multinucleate stage of *Creolimax* is not characterized by a tight regulation of adhesive, signaling, and TF activities.

## Differential alternative splicing down-regulates specific pathways in *Creolimax*

Alternative splicing is a widespread mechanism for post-transcriptional gene regulation in animals and many other eukaryotes. To evaluate the extent to which alternative splicing expands transcriptome complexity in the intron-rich genome of *Creolimax* (6.5 introns per gene), we undertook a comprehensive analysis of intron retention and exon skipping (ES) events, and quantified their inclusion levels using previously developed pipelines (*Braunschweig et al., 2014*; *Irimia et al., 2014*) (see Material and methods). We observed 3927 intron-retention events that affected 2172 genes. In contrast, we found only 211 genes affected by ES. Despite having a two-fold higher intron density and longer introns compared to *Capsaspora*, both holozoan species show comparable alternative splicing profiles that are highly dominated by intron retention (*Sebé-Pedrós et al., 2013*). This provides further support for the hypothesis that ES-dominated alternative splicing is a unique feature of animal transcriptomes (*Sebé-Pedrós et al., 2013*; *Mcguire et al., 2008*; *Irimia and Roy, 2014*).

Next, we evaluated the extent to which intron retention is differentially regulated across *Creolimax* stages. We found 865 introns with differential retention between stages (differential percent intron retention (ΔPIR) >15), the majority of which were more retained in the amoeboid stage (*Figure 3A*). Reverse transcription-polymerase chain reaction (RT-PCR) assays confirmed differential retention for all 10 tested introns (*Figure 3D* and *Figure 3—figure supplement 2*). Despite these marked differences in intron retention between stages, the size distribution of retained introns was similar in both stages, and comparable to that of constitutively spliced introns (PIR<2% in all

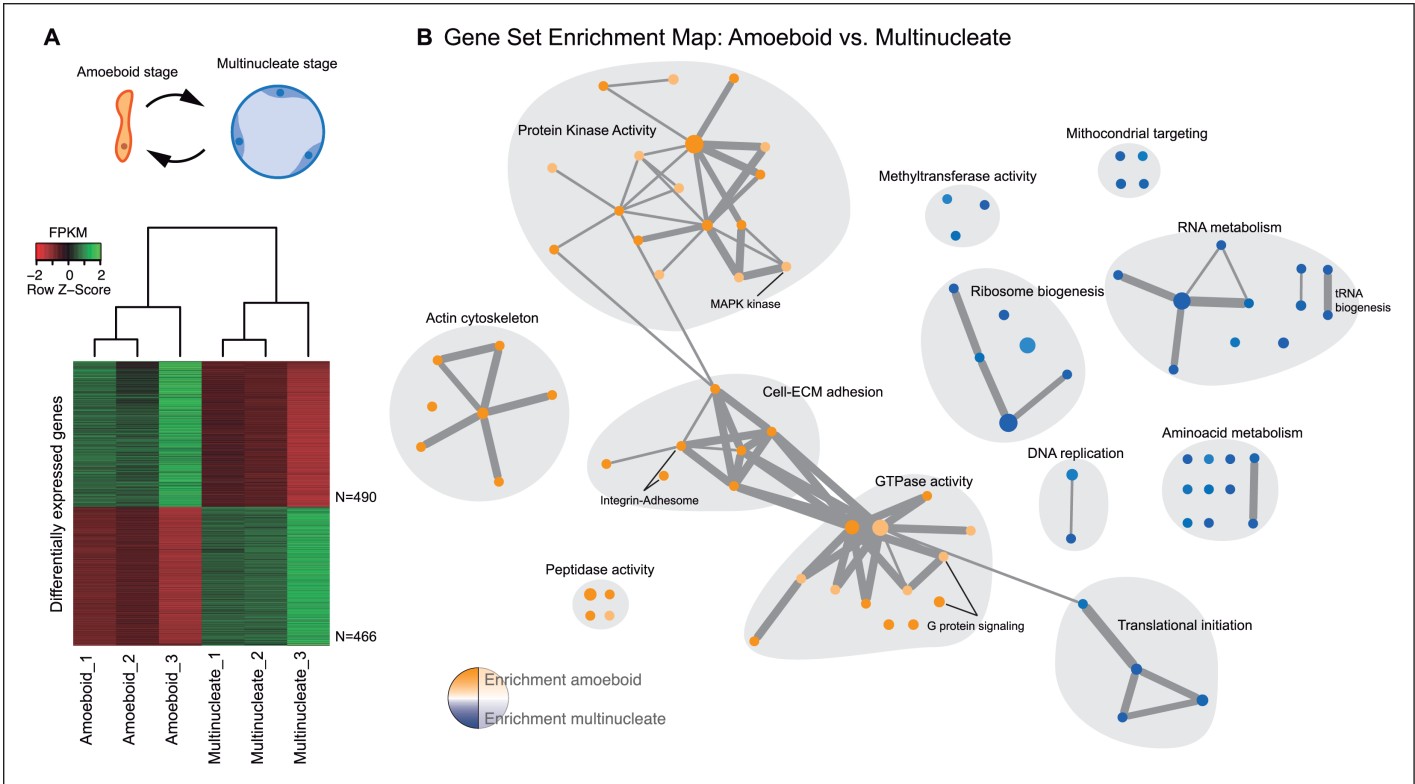

**Figure 2.** Differential gene expression in *Creolimax*. (A) Diagram of the amoeboid and multinucleate stages, and heatmap showing the significantly differentially expressed genes across biological replicates in the pair-wise stage comparison. (B) Gene set enrichment analysis for the two stages. Orange represents enrichment in the amoeboid stage and blue represents enrichment in the multinucleate stage, color intensity depicts level of significance (p value). Node size represents the total number of genes in each GO, and edge width represents the total number of genes shared between each enriched GO category. Functionally related GOs are manually circled in gray shade according to functional and genic redundancy established by network connectivity. Complete list of GOs and inclusive groupings are found in *Figure 2—source data 1*. GOs, Gene Ontologies.

The following source data and figure supplements are available for figure 2:

**Source data 1.** GOs enrichments and groupings from *Figure 2B*.

**Figure supplement 1.** Pfam domain set enrichments in differentially expressed genes.

**Figure supplement 2.** Differential TFome expression.

samples) (*Figure 3—figure supplement 1A,B*). GO enrichment analysis of genes with amoeboid-specific intron retention revealed enrichment in spindle pole formation and other mitosis-related activities (*Figure 3C*), supporting an active role of IR in down-regulating these functions in this stage. Indeed, consistent with recent reports in vertebrates (*Braunschweig et al., 2014*), genes with amoeboid-specific intron retention showed significantly lower steady-state mRNA levels compared to the multinucleate stage (*Figure 3—figure supplement 1C*; p<0.0001, Wilcoxon signed rank test; the converse was true for genes with multinucleate-specific intron retention). Therefore, intron retention is a conserved mechanism for reducing transcript levels in pathway-specific genes from unicellular holozoans to vertebrates.

In the case of ES, we found 63 exons with differential inclusion levels across stages (*Figure 3B*). RT-PCR assays were used to validate the differential inclusion for all 12 tested cases (*Figure 3E* and *Figure 3—figure supplement 2*). Only a minority of these exons (*Torruella et al., 2015*) keep the reading frame upon skipping, and only two of these overlapped with functional Pfam domains, suggesting that, similar to intron retention, ES in *Creolimax* predominantly contributed to functionally down-regulating specific genes. GO enrichment analysis revealed that differentially spliced exons

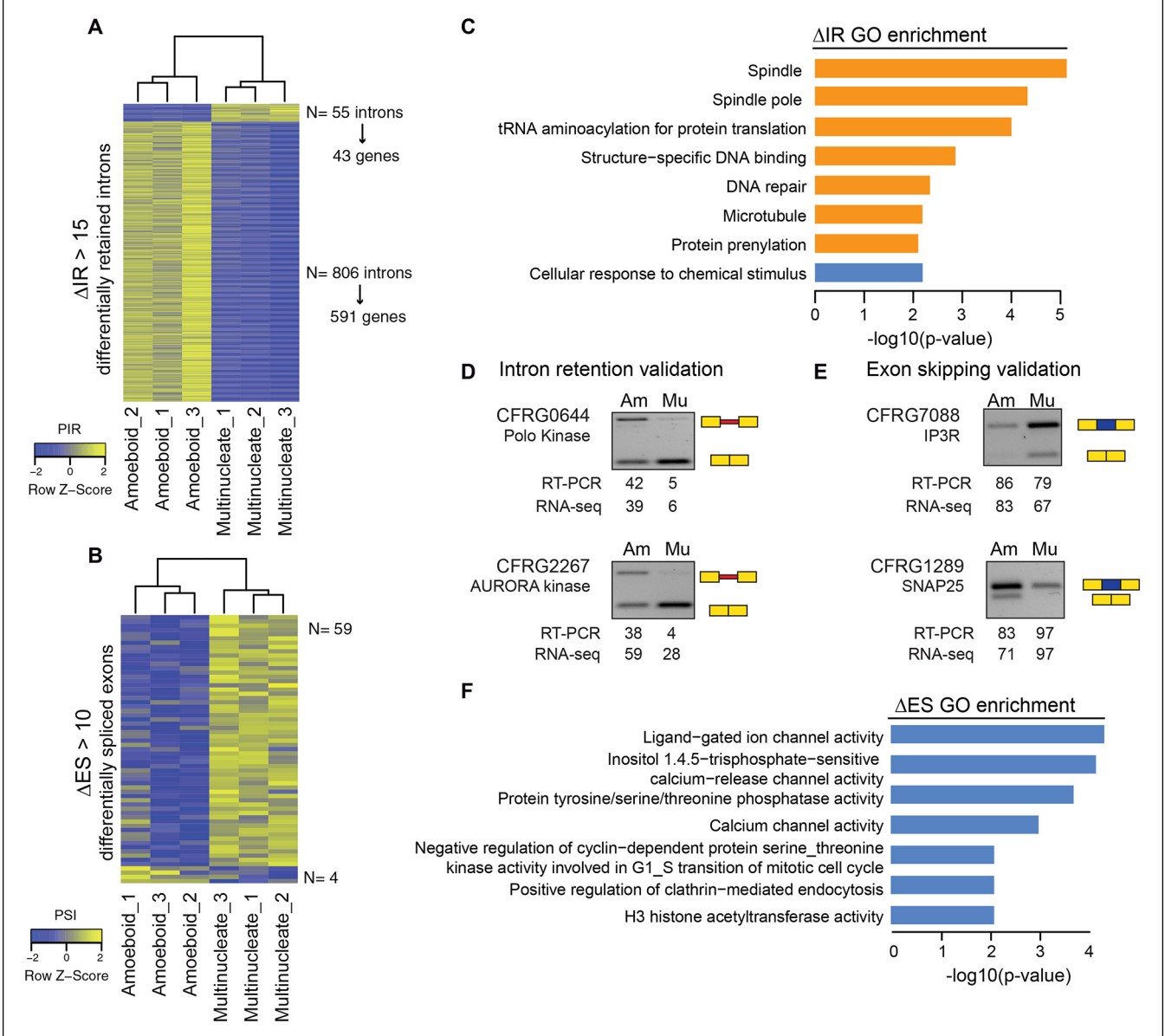

**Figure 3.** Regulated alternative splicing modes in *Creolimax*. (**A**) Heatmap showing PIR inclusion levels of differentially retained introns. (**B**) Heatmap showing the PSI levels of differentially skipped exons. (**C**) GO enrichment activities of the genes showing differential IR. Bar length indicates the significance of the enrichment, orange indicates those with higher inclusion levels in the amoeboid stage and blue those with higher inclusion levels in the multinucleate stage. (**D–E**) RT-PCR validations of selected IR and ES events. The values correspond to relative intensity of the alternative isoform (retained intron or skipped exon) bands in the RT-PCR and the proportions observed for the inclusion values in the RNA-seq. (**F**) GOs enrichment of genes with differential ES, in blue those with higher exon inclusion levels in the multinucleate stage. ES, exon skipping; GO, Gene Ontology; IR, intron retention; PIR, percent intron retention; PSI, percent spliced in; RT-PCR, reverse transcription-polymerase chain reaction; RNA-seq, RNA sequencing.

The following figure supplements are available for figure 3:

**Figure supplement 1.** Intron size and transcriptional levels of differentially retained introns.

**Figure supplement 2.** Validations of intron retention and ES events.

belong to genes involved in various biological processes, including channel activity and histone modifications (*Figure 3F*).

## A population of lincRNAs with regulated expression in *Creolimax*

Another layer of transcriptional complexity is provided by long non-coding RNAs, which are increasingly recognized as important players in animal development and cell type-specific genome regulation (*Ulitsky and Bartel, 2013*; *Morris and Mattick, 2014*; *Sauvageau et al., 2013*). To characterize the repertoire of long non-coding RNAs in *Creolimax*, we assembled a de novo transcriptome from the RNA-seq data. We filtered out transcripts shorter than 200 bp and mapped the rest to the genome. We then applied a pipeline to identify putative non-coding RNAs, including searches for coding potential, homology, untranslated region (UTR) mis-annotation, and low expression (see Material and methods). Restricting our search to transcripts that did not overlap with protein-coding genes, known as long-intergenic RNAs (lincRNAs ), we identified 692 putative lincRNA loci in *Creolimax*. In comparison to protein-coding transcripts, lincRNAs in *Creolimax* were shorter in length, harbored fewer exons, had longer exons, and had a lower GC content (*Figure 4—figure supplement 1*). Moreover, overall transcription levels of lincRNAs were significantly lower than those of protein-coding genes (p<0.01, Wilcoxon rank sum test). Interestingly, all those characteristics have been reported for animal lincRNAs (*Hezroni et al., 2015*; *Pauli et al., 2012*; Gaiti et al., 2015).

To infer possible functions of *Creolimax* lincRNAs, we first looked at the functional annotations of the closest neighboring protein-coding genes, as animal lincRNAs are enriched near developmental genes and TFs (*Ulitsky and Bartel, 2013*). In *Creolimax*, the only significantly enriched GOs of lincRNA closest neighboring genes were metabolic activities (p<0.01). However, when we analyzed the vicinity of all the TFs, we found that 23.6% had at least one neighboring lincRNA, a significant enrichment compared to the rest of the genome (14.5%, p=0.0007, Fisher's exact test). Next, we searched for putative homologs of the lincRNAs in the transcriptomes of closely related species and other unicellular holozoans, but we did not retrieve any positive hits. This pattern of rapid evolutionary sequence turnover of lincRNAs has also been described for animals (*Gaiti et al., 2015*; *Kapusta and Feschotte, 2014*), where homology detection based on sequence similarity is restricted to short evolutionary distances.

In animals, expression of lincRNAs is generally restricted to specific tissues and organs (*Ulitsky and Bartel, 2013*; *Necsulea and Kaessmann, 2014*). In *Creolimax* we detected 51 lincRNA loci that were differentially expressed between the amoeba and the multinucleate stage (*Figure 4*). Overall, only 7% of the total detected lincRNAs are differentially expressed, compared to 10% of coding genes (p=0.0059, Fisher's exact test). Thus, in stark contrast to animals, the lincRNAs in *Creolimax* appear to be less cell type-specific than coding genes.

To test whether the transcription of lincRNAs is linked to their upstream genes, we first subdivided lincRNAs into two categories: head-to-head, where the lincRNA and the adjacent protein-coding gene have opposite orientation and may share the same promoter, and head-to-tail, where the lincRNA promoter is downstream of the adjacent protein-coding gene (*Figure 4—figure supplement 2*). Head-to-head orientation was significantly over-represented in the genome (485/692, 70% of the total vs the 50% expected by chance, $p < 3.6e^{-14}$, $\chi^2$ test). Furthermore, pairs in this head-to-head orientation showed a higher proportion of positively correlated expression levels, although this was similar to pairs of protein-coding genes in the head-to-tail orientation. Moreover, 80% of differentially regulated lincRNAs are in head-to-tail orientation or display uncoupled transcriptional profiles in relation to their neighbor genes (p<0.05 Pearson's correlation coefficient). Although the possibility that bidirectional promoters are the main source of lincRNAs in *Creolimax* cannot be excluded, differential regulation of lincRNA is largely independent of their neighboring protein-coding genes.

Finally, we analyzed the impact of alternative splicing in lincRNAs. Although the majority of lincRNAs have a single exon, 132 (19.1%) loci have two or more exons. Among those multi-exon transcripts, we detected 183 intron-retention events (*Figure 4E*). Of these, eight events differed between samples (PIR>15), suggesting differential regulation of lincRNAs also at the level of splicing. Therefore, both transcriptional regulation and alternative splicing may play a role in the active regulation of lincRNAs in *Creolimax*.

## Evidence of species-specific cell type evolution in Holozoa

Direct comparison of steady-state transcript abundance is useful to reveal cross-species molecular homologies in animals, as homologous tissues in different species show more similarity in the

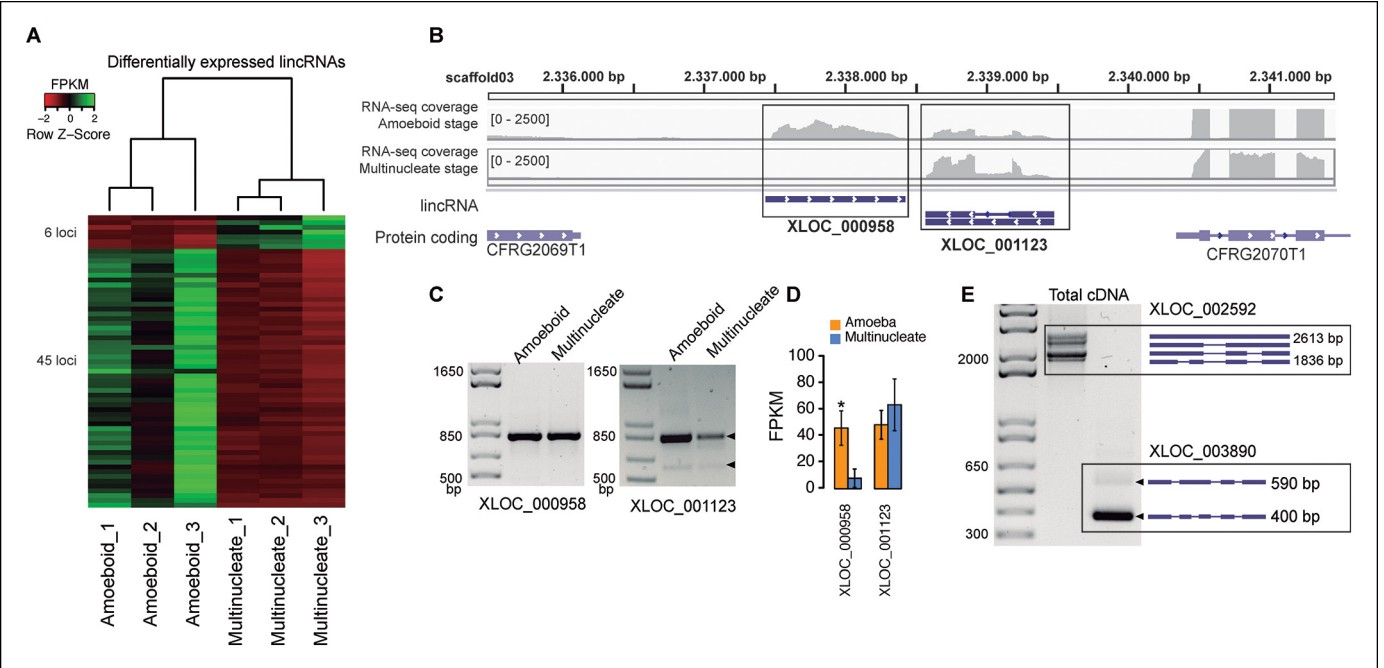

**Figure 4.** Transcriptional and post-transcriptional regulation of lincRNAs in *Creolimax*. (A) Heatmap showing transcriptional levels of significantly differentially expressed lincRNAs across biological replicates of amoeboid and multinucleate stages. (B) Example of genomic region where two lincRNA loci are found in tail-to-tail orientation surrounded by two protein-coding genes. (C) RT-PCR validations of the lincRNA loci. (D) Barplot depicting the average gene expression of those lincRNA in each stage. (E) Alternative splicing isoforms of lincRNAs showing various degrees of IR. IR, intron retention; lincRNA, long intergenic non-coding RNAs; RT-PCR, reverse transcription-polymerase chain reaction.

The following figure supplements are available for figure 4:

**Figure supplement 1.** Genomic architecture of lincRNAs compared to protein-coding genes.

**Figure supplement 2.** Gene orientation and transcriptional co-regulation of neighboring genes.

transcriptional profiles than do non-homologous tissues in a single species (*Barbosa-Morais et al., 2012*; *Chan et al., 2009*; *Brawand et al., 2011*). Using a similar rationale, we sought to compare the different cell stages of unicellular holozoans. From normalized transcriptional levels of a set of 2177 one-to-one protein-coding orthologs between *Salpingoeca*, *Capsaspora*, and *Creolimax*, we calculated pairwise Spearman correlation distances, which were then used to perform hierarchical clustering and neighbor-joining tree reconstructions (*Figure 5*). Both approaches showed consistent trees with species-specific clustering for the different stages. The signal obtained from those 2177 genes was enough to cluster samples consistent with their whole transcriptome patterns (except one sample from *Salpingoeca*).

Direct cell type comparisons between species revealed that *Creolimax* multinucleate stage and *Capsaspora* cystic stage are the most dissimilar among all samples, whereas the amoeboid stage in *Creolimax* has a stronger correlation with the filopodial and aggregative stages of *Capsaspora* (*Figure 5A*). In order to characterize gene sets responsible for those cross-species similarities, we used a principal component analysis (PCA) to distinguish between the species-specific factors and the underlying biological similarities. Species-specific variability was captured by PC1 (47.96%); however, when we plotted PC2 and PC3 we obtained different groupings of cell types independent of their species of origin (*Figure 5C*). PC2 grouped the multinucleate stages of *Creolimax* with the aggregative and the filopodial stages of *Capsaspora*. This grouping was explained by top-loading genes involved in DNA replication and spliceosomal activities (*Figure 5D*), a pattern consistent with the down-regulation of mitotic activity and cell growth in the *Capsaspora* cystic stage (*Sebé-Pedrós et al., 2013*). On the other hand, PC3 grouped the amoeboid stage of *Creolimax* and the filopodial and aggregative stages of *Capsaspora*. PC3 was loaded with genes involved in the integrin

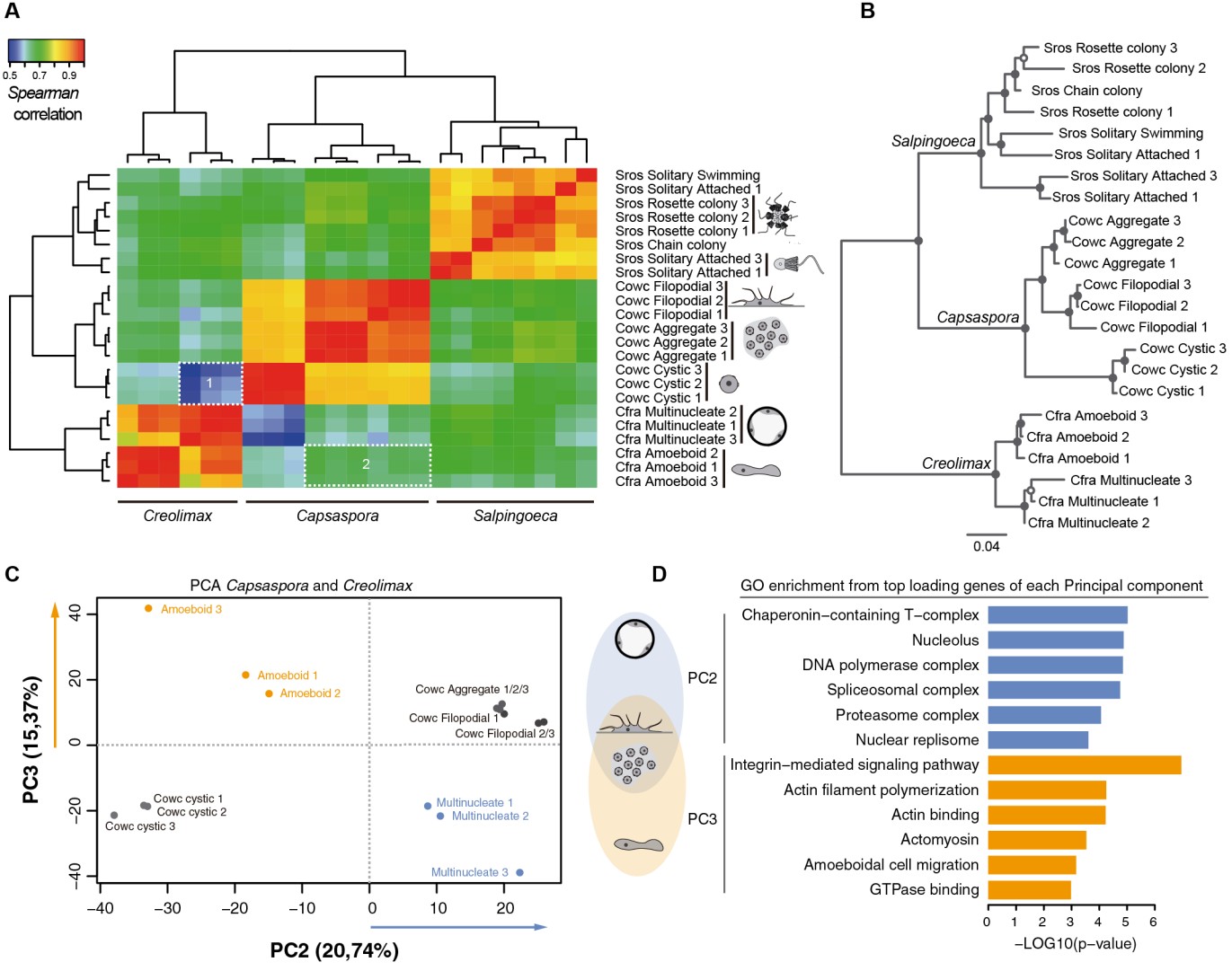

**Figure 5.** Holozoan cross-species comparison of transcriptional profiles. (**A**) Symmetrical heatmap of the pair-wise Spearman correlation coefficients for the gene expression profiles of each cell stage. For each sample, log2(cRPKM+1) expression levels were obtained for 2177 one-to-one orthologs in the three species analyzed (see Materials and methods). Dashed-line squares highlight the direct comparisons for 1) Cfra multinucleate stage replicates against Cowc cystic stage replicates and 2) Cfra amoeboid stage replicates against Cowc aggregate and filopodial stage replicates. (**B**) Neighbor-joining tree of the species cell stages based on the aforementioned Spearman correlation distances matrix. Filled circles represent >95% bootstrap replicate nodal support. (**C**) The cell types plotted according to the values of the principal components 2 and 3 from a PCA of a dataset of 3030 1-to-1 orthologs between *Capsaspora* and *Creolimax*. (**D**) The significant GO enrichments for the top positive loading genes (>0.03) of the principal component 2 and 3. Cfra, *Creolimax fragrantissima*; Cowc, *Capsaspora owczarzaki*; GO, Gene Ontology; Sros, *Salpingoeca rosetta*; PCA, principal component analysis.

adhesome and amoeboid actin-based motility and signaling activities (*Figure 5D*). Therefore, despite the vast phylogenetic divergence among holozoan species, some functional patterns can be recovered through direct comparison of transcriptional profiles.

We used the same approach to compare RNA-seq data from a wide range of human cell types and tissues with those of *Creolimax* (*Figure 6A*). Certain human cell types showed differential grades of similarity in the expression profiles with each of the *Creolimax* stages. The human cell types with a higher positive correlation with the multinucleate stage were those with high proliferation rates, including embryonic stem cells, induced pluripotent stem cells and transformed cell lines (293T, HeLA, K562) (*Figure 6B*). Similar patterns were obtained using PCA on the normalized steady-state transcriptional levels from both species. PC1 explained most of the variability due to species-specific transcriptomic profiles, whereas PC2 distinguished between highly proliferative cell types and the

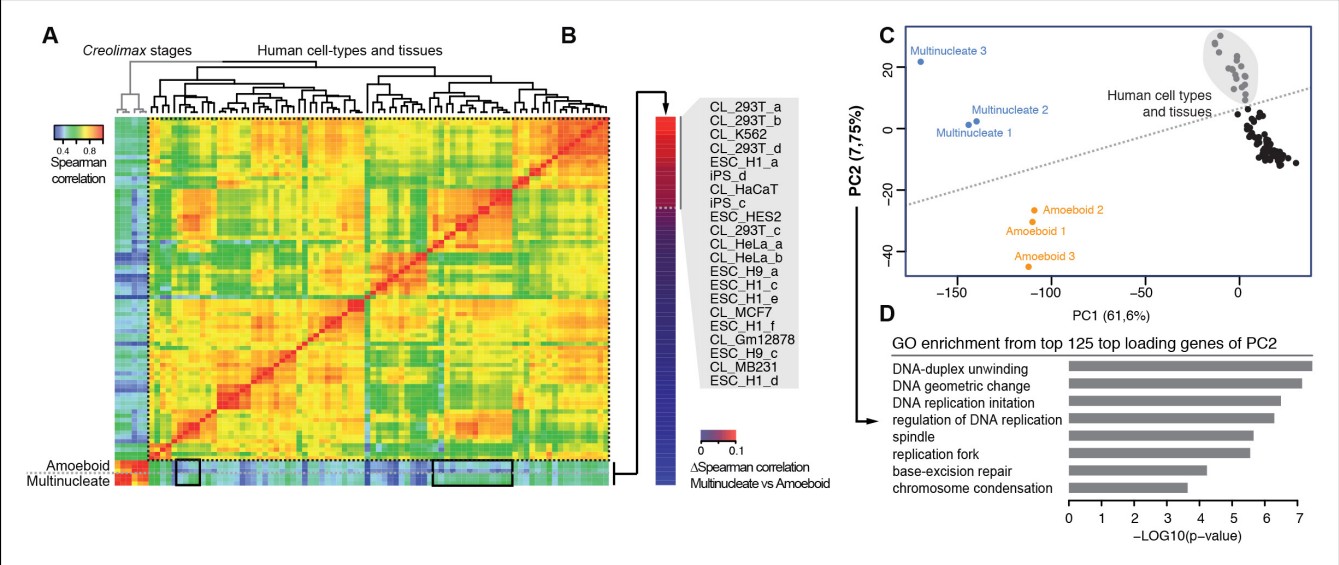

**Figure 6.** Comparison of human and *Creolimax* cell types and tissues. (**A**) Symmetrical heatmap of the pair-wise Spearman correlation coefficients for the gene expression profiles of each cell type or tissue. For each sample log2(FPKM+1) expression levels were obtained for 2272 one-to-one orthologs between *Creolimax* and human (see Materials and methods). (**B**) The human cell types sorted by the difference of Spearman correlation between the amoeboid and the multinucleate cell stages. Highlighted in gray are those that displayed the major differences (>0.05). (**C**) The cell types plotted according to values of the principal components 1 and 2 from a PCA of the same transcriptional dataset of 2272 orthologs. The dots in gray are the human cell lines highlighted in the previous section. (**D**) The significant GO enrichments for the top positive loading genes of the principal component 2 (>0.04). Sampled human cell types described in *Figure 6—source data 1*. GO, Gene Ontology; PCA, principal component analysis.

The following source data is available for figure 6:

**Source data 1.** Human RNA-seq datasets used in this analysis.

rest in both species (*Figure 6C*). Consistent with this observation, GO enrichment analysis of the top 125 genes that most contributed to PC2 showed highly significant enrichment for genes involved in genome replication and proliferation (*Figure 6D*). These results suggest that a signal from the evolutionarily conserved machinery for cell proliferation in eukaryotes (*Harashima et al., 2013*; *Cross et al., 2011*) can be detected from direct expression pattern comparisons between the multinucleate stage of *Creolimax* and highly proliferative human cell types.

## Conserved ancient co-regulated gene modules: the Integrin adhesome

To obtain more specific insights into the evolution of co-regulated gene programs across holozoans, we investigated several gene modules with key roles in animal multicellularity. Despite gene co-regulation providing indirect evidence for shared functionality in a given species (*Gerstein et al., 2014*; *Stuart, 2003*), comparative analysis of transcriptional co-regulated gene modules in different species may offer additional insights into the functional evolution of the animal multicellular toolkit.

First, we focused on the integrin adhesome, which is crucial for ECM adhesion in animals. The core components of the integrin adhesome have been identified in *Capsaspora* (*Sebe-Pedros et al., 2010*). Interestingly, these components are significantly up-regulated in the aggregative stage (*Sebé-Pedrós et al., 2013*). In contrast, most integrin adhesome components have been lost in choanoflagellates (*Sebe-Pedros et al., 2010*). In the case of *Creolimax*, we found a nearly complete repertoire of the integrin adhesome components (*Figure 7D*), and most of them are significantly up-regulated in the amoeboid stage (*Figure 7A*). To test whether the integrin adhesome components constitute a co-regulated gene module in the three unicellular holozoans (ichthyosporeans, filastereans, and choanoflagellates), we calculated the transcriptional Pearson correlation coefficients between all of the genes encoding the integrin adhesome for each species (*Figure 7A–C*). In the case of *Creolimax*, we observed a remarkable co-regulation among all of the components. The only exception is the Src tyrosine kinase (*Figure 7A*), suggesting that tyrosine kinase signaling is not

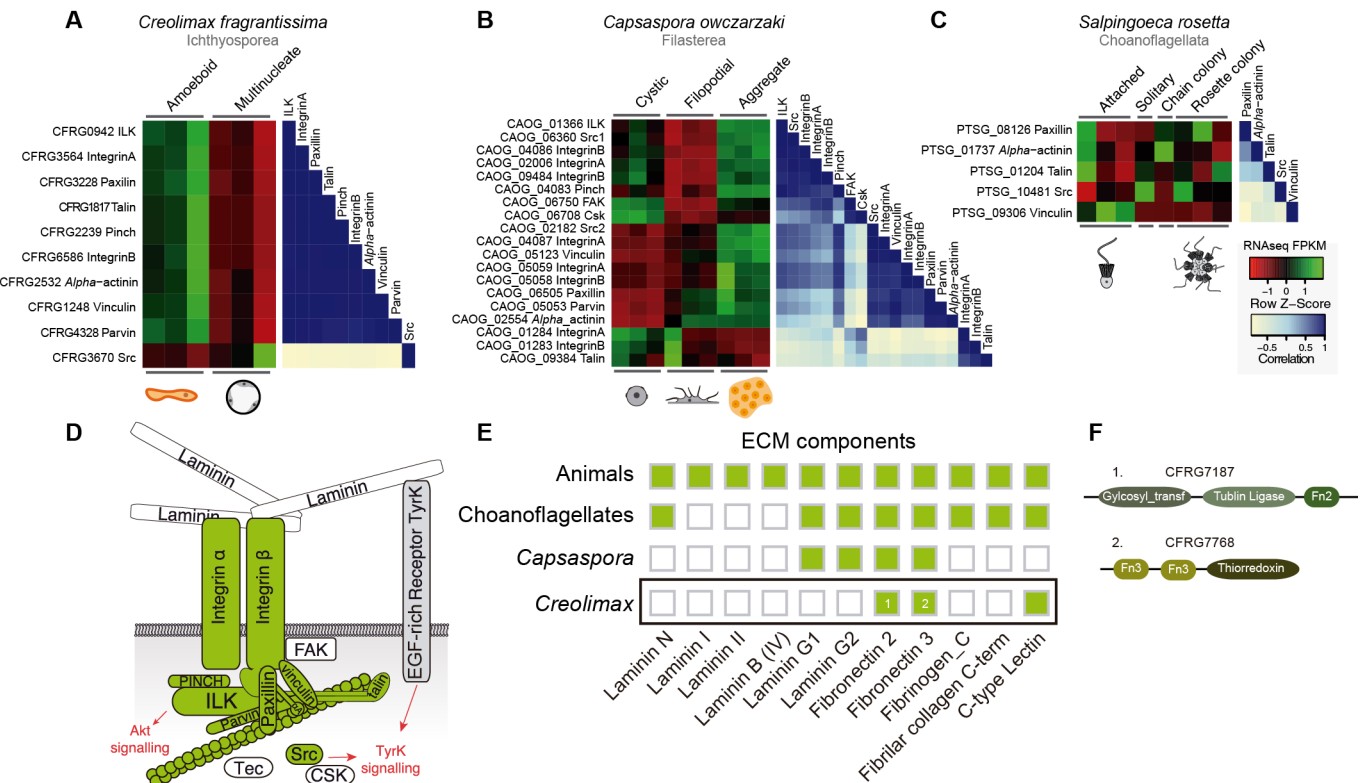

**Figure 7.** Co-regulation of the integrin adhesome in holozoans. Heatmaps depicting expression levels of integrin adhesome orthologs (red–green) and their pair-wise Pearson correlation coefficients (white–blue) obtained from genome-wide FPKM transcriptional levels in the ichthyosporean *Creolimax* (**A**), the filasterean *Capsaspora* (**B**), and the choanoflagellate *Salpingoeca* (**C**). (**D**) Diagram of integrin adhesome components, those in green are found in *Creolimax* and those in white are absent. In gray, a tyrosine kinase receptor with extracellular EGF domains encoded *Creolimax* genome that could be interacting with an ECM component. (**E**) Repertoire of animal ECM domains in the three unicellular holozoan genomes; green = presence, white = absence. (**F**) Pfam domain architectures of fibronectin-domain containing genes in *Creolimax*. ECM, extracellular matrix; FPKM, fragments per kilobase of exon per million fragments mapped.

The following figure supplements are available for figure 7:

**Figure supplement 1.** Co-regulation of the filopodial molecular toolkit genes in holozoans.

**Figure supplement 2.** Co-regulation of the pre- and post- synaptic genes in holozoans.

connected to the integrin signaling pathway in *Creolimax*, consistent with the absence of a focal adhesion kinase in the ichthyosporean lineage (**Suga et al., 2014**). In the filasterean *Capsaspora*, we identified a more complex pattern of co-regulation, grouped in two submodules that are associated with specific paralogs of both *alpha* and *beta* integrins (**Figure 7B**). This complex pattern of co-regulation could be affected by the number of *Capsaspora*'s samples, higher than those analysed in *Creolimax*. In contrast, we did not detect any co-regulation among the few conserved integrin adhesome components in the choanoflagellate *Salpingoeca* (**Figure 7C**), suggesting that gene loss in the choanoflagellate lineage was accompanied by dismantling of this ancient co-regulatory module.

In animals, integrins mediate cell-adhesion through binding to ECM proteins, such as laminins and fibronectins (**Cromar et al., 2014**). In *Capsaspora*, laminin-containing genes with predicted secretion signals are up-regulated in the aggregative stage (**Sebé-Pedrós et al., 2013**). However, despite showing a co-regulated integrin adhesome, we did not find any laminin-containing genes in *Creolimax* (**Figure 7E**). The only ECM-related domains that we found in the genome were fibronectins, but those are fused to metabolic domains unrelated to the ECM and none had a secretion

signal peptide (*Figure 7F*). Therefore, it seems unlikely that integrins are involved in adhesion to an endogenous animal-like ECM in *Creolimax*.

Next, we investigated the filopodial machinery (*Sebé-Pedrós et al., 2013*). We observed strong positive co-regulation of a core set of components in all unicellular holozoans (*Figure 7—figure supplement 1*), consistent with the use of an ancestral cellular machinery for filopodial formation. In stark contrast to the situation for cell replication machinery, we did not observe a single co-regulated module in any unicellular species for genes involved in animal neuronal pre-synaptic and post-synaptic processes, including most pairs of genes that are known to directly interact in animals (with the exception of syntaxin, synaptobrevin, and synaptogryin, which are involved in secretory vesicle formation (*Burkhardt et al., 2011*); *Figure 7—figure supplement 2*). In fact, a lack of interaction has been shown for some of the core proteins involved in the post-synaptic scaffold in *Salpingoeca* (*Burkhardt et al., 2014*). Thus, our results suggest that some molecular complexes directly involved in cell morphology and behavior already formed co-regulated gene modules in unicellular holozoans, whereas other complexes involved in unique animal cell types were assembled later, despite having conserved orthologs.

## The secretome of *Creolimax* shows convergent adaptations to a specialized osmotrophic lifestyle

Ichthyosporeans are an interesting case of convergent evolution of fungal-like traits; in fact, they were once thought to be fungi based on their lifestyle and morphology (*Glockling et al., 2013*; *Mendoza et al., 2002*). For instance, both groups have a cell wall, similar parasitic lifestyles, and a specialized osmotrophic feeding mode, unlike any of the other holozoan lineages (*Figure 1*). Specialized osmotrophs are characterized by their highly adapted secretomes that are key to the external digestion of complex compounds. Therefore, we analyzed the secretome of *Creolimax* to investigate the convergent evolution of this specialized osmotrophic feeding mode in the ichthyosporeans. In addition, secretome analysis should provide further insights into the production and modification of ECM and signaling ligands.

To characterize the secretome, we performed a combination of in vivo and in silico approaches. First, using high-throughput proteomics of an in vivo secretome sample in culture conditions, we identified 91 proteins. Next, we applied an in silico approach to predict 453 proteins that are likely to be secreted (see Material and methods). Interestingly, only 48 proteins were common to both datasets, indicating that proteins without a canonical signal peptide or with additional transmembrane domains could also be found in the in vivo secretome (*Figure 8A*). Among the 43 non-canonically secreted proteins, we detected some with transmembrane domains that could be the product of shedding, as proposed for other species (*Meijer et al., 2014*). For instance, we found peptides corresponding to the extracellular region of both *alpha* and *beta* integrins. Although shedding of integrins has been observed during the inflammatory response in animals (*Gomez et al., 2012*), the functional implications of integrin shedding in *Creolimax* remain elusive.

Both secretome datasets were highly enriched in peptidase and proteolytic activities (*Figure 8B*), whereas only the in silico dataset showed further enrichments in other catabolic functions such as lipase and hydrolase activities. Analysis of Pfam-domain enrichment consistently revealed proteolytic and peptidase domains in both datasets (*Figure 8C*). In contrast, an in-depth search for candidate ECM-related proteins and diffusible ligands based on Pfam domains (using known animal domains as EGF, DSL, or laminins) retrieved no positive hits. Thus, the proteolytic activity of the *Creolimax* secretome does not seem to be related to modifying the endogenous ECM, but possibly to the adaptation to a specialized osmotrophic feeding mode.

Osmotrophic lifestyles are characterized by the external digestion of complex polymers (*Talbot, 2013*). Therefore, the enrichment in proteolytic activities observed in the secretome strongly suggests that proteins and peptides are the main food source of *Creolimax*, at least under culture conditions. Such external digestion requires a coupled mechanism for nutrient uptake (*Talbot, 2013*). Consistent with this requirement, we found 38 genes with four distinct Pfam domains (PF00324, PF01490, PF03169 and PF00854) that are predicted to be involved in the amino acid and oligopeptide transporter activity.

The most-abundant gene family in *Creolimax* secretome was the trypsins (15 proteins in the in vivo proteomic dataset out of the 31 genes found in the genome). Trypsins are usually found in multiple copies in animal genomes, and they have important roles in the digestive system as serine

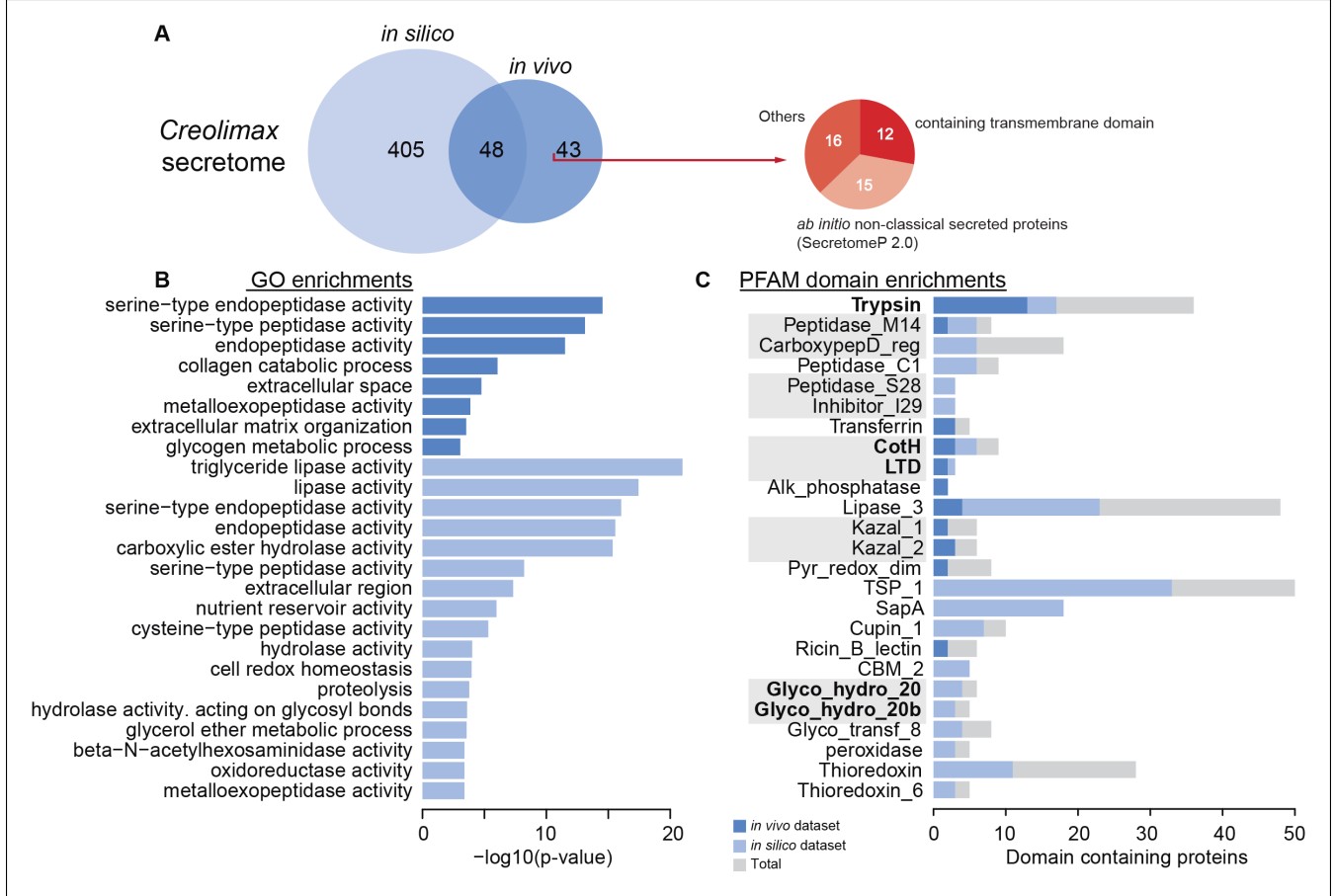

**Figure 8.** Functional enrichments of *Creolimax* secretome. (**A**) Venn diagram showing the number of genes identified in the *Creolimax* secretome through an in silico approach (see Material and methods) and an in vivo proteomics approach. A circle diagram describes the features of genes only identified in the in vivo approach, lacking a signalP or having TM domains . (**B**) GO categories and (**C**) PFAM domains enriched in the secretome; in dark blue are those enriched in the in vivo dataset; in pale blue are those enriched in the in silico dataset; in gray are the total amount of PFAM-domain containing genes in the genome. GO, Gene Ontology.

The following source data is available for figure 8:

**Source data 1.** In vivo proteomics of *Creolimax* secretome.

proteases and also as ECM remodellers. To further complement these observations, we profiled a wide range of eukaryotes and found independent expansions of trypsins in other osmotrophic lineages (*Figure 9A*). Among these, two fungal species (*Coemansia reversa* and *Conidiolobus coronatus*) and an oomycete (*Saprolegnia parasitica*) are also animal-dwelling parasites, thus sharing a similar lifestyle to the known ichthyosporeans. Phylogenetic analyses of *Creolimax* trypsins (*Figure 9B*) revealed that most are the product of rapid lineage-specific gene duplications, a common source of molecular adaptation (*Kondrashov, 2012*). Moreover, we found very distinct patterns of transcript and protein abundance (measured by the number of unique peptides identified) across the trypsin paralogs (*Figure 9B*), suggesting a rapid process of functional diversification. Thus, trypsins seem to be a recurrently used effector gene family in osmotrophic eukaryotes, and have evolved rapidly in *Creolimax*.

## Recurrent prokaryotic LGT events shaped the *Creolimax* secretome

LGT is a rare but recurrent mechanism for the acquisition of osmotrophy-related genes in both fungi and oomycetes (*Talbot, 2013*). We therefore investigated LGT in *Creolimax* by building automatic taxon-rich phylogenies. Manual inspection of these trees yielded a list of 163 genes that confidently

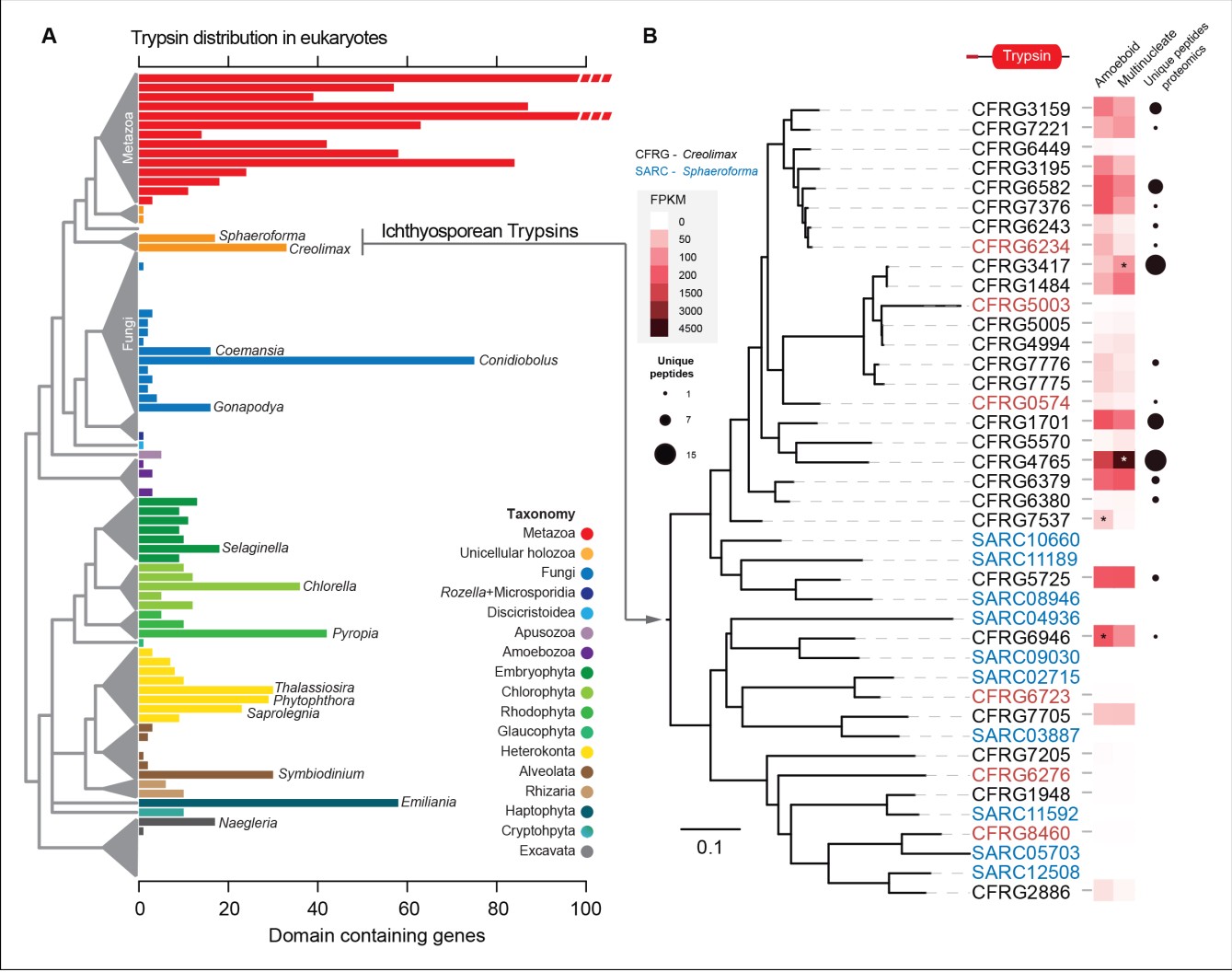

**Figure 9.** Trypsin evolution. (**A**) Barplot showing the total number of Trypsin proteins (PF00089) found in the genomes of diverse eukaryotes. Branches are color coded according to the taxonomy shown in the legend. (**B**) Maximum-likelihood phylogenetic tree based on the amino acid sequence of the Trypsin domain from *Creolimax fragrantissima* and *Sphaeroforma arctica*. Expression levels obtained from genome-wide FPKM calculation are shown. Number of unique peptides obtained from the in vivo secretome proteomic dataset is also shown. In red are those genes that do not present a signal peptide according to SignalP. FPKM, fragments per kilobase of exon per million fragments mapped.

branched within bacterial clades, supporting bacterial origins. Of these, 35 (21.5%) were found only in *Creolimax* among all eukaryotes sampled, 50 (30.7%) were shared with the close relative *Sphaero-forma arctica*, and 71 (43.6%) were shared with other ichthyosporeans (***Figure 10—figure supplement 2***). Although most LGT genes (102 out of 163) were intron-less, the vast majority (143, 87.7%) showed transcriptional support from our polyA-selected RNA-seq, minimizing the possibility of bacterial contaminations. LGT from bacteria accounts for 1.8% of the total proteome, which is slightly higher than in other eukaryotic genomes (***Alsmark et al., 2013***). Importantly, we found 6 genes acquired by LGT in the in vivo secretome and 18 more in the in silico secretome (***Figure 10—figure supplement 2***). Therefore, similar to other eukaryotic lineages (***Richards et al., 2011***; ***Richards et al., 2011***), ichthyosporeans have enriched their osmotrophy-related gene complement through prokaryotic LGT.

Among the six genes of prokaryotic origin that we found in the in vivo secretome, three belonged to the spore coat homology (CotH) family (PF08757). CotH proteins were first described as being fundamental for spore coat formation in the bacteria *Bacillus subtilis* (***Naclerio et al., 1996***), but they have recently been characterized as critical factors for host invasion in the fungus *Rhizopus*

*oryzae* (*Gebremariam et al., 2014*). Our phylogenetic reconstruction of CotH family evolution revealed that the presence of CotH homologs in *R. oryzae* and other fungi (only belonging to the class mucorales) originated from an LGT event that was independent of those found in ichthyosporeans (*Figure 10* and *Figure 10—figure supplement 1*). Interestingly, both lineages have expanded their CotH family members after LGT acquisition. Moreover, CotH genes were also found in other eukaryotic and archaeal lineages, suggesting a complex history of interdomain LGT similar to that recently described for other gene families (*Funkhouser-Jones et al., 2014*; *Chou et al., 2014*). Active gene duplication and domain shuffling characterized the evolutionary history of CotH in the ichthyosporea, where we could observe variable transcriptional levels and peptides among distinct paralogs, as well as the acquisition of an N-terminal LTD domain (PF00932) (*Figure 10B*). Active transcription and secretion of *Creolimax* CotH genes in axenic culture conditions underscore their putative role in host invasion and highlight the importance of LGT in effector gene acquisition across different osmotrophic lineages.

## Discussion

In this study, we showed how different layers of genome regulation shape the coenocytic development and lifecycle of the ichthyosporean *Creolimax*, a unicellular relative of animals. These regulatory layers include complex mechanisms that also play a role in animal development and cellular differentiation. In *Creolimax*, differential genome regulation is not only limited to the transcriptional control of protein coding genes, but also includes cell-stage specific alternative splicing and lincRNA

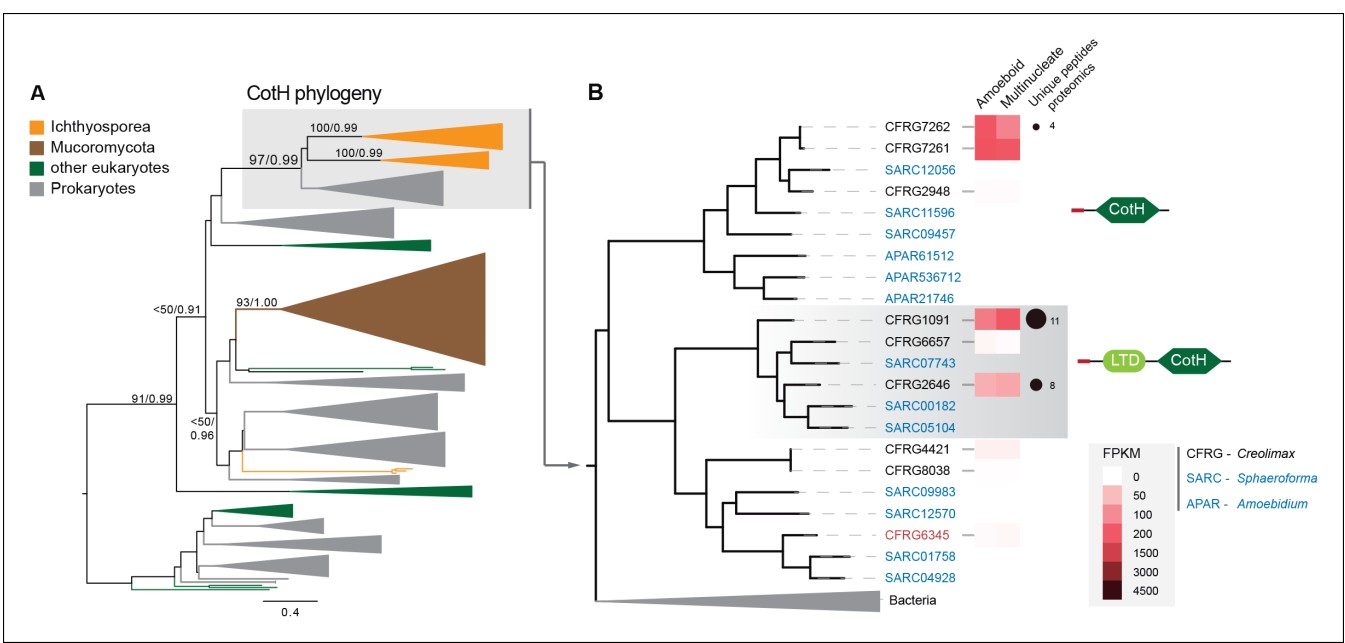

**Figure 10.** CotH evolution. (**A**) Maximum-likelihood phylogenetic tree of the CotH domain (PF08757). Nodal support is shown in key branches (100 maximum likelihood replicates bootstrap values and Bayesian posterior probabilities). Color code indicates taxon distribution in each clade as depicted in the legend; for a detailed tree, see *Figure 10—figure supplement 1*. (**B**) Detail of the phylogenetic tree depicting ichthyosporean CotH sequences, covering *Creolimax, Sphaeroforma arctica*, and *Amoebidium parasiticum*. Expression levels obtained from genome-wide FPKM calculation and the number of unique peptides obtained from the in vivo secretome proteomic dataset are shown. Domain configurations obtained from a PfamScan analysis. Gene identifiers in red are those that do not present a signal peptide according to SignalP. FPKM, fragments per kilobase of exon per million fragments mapped.

The following figure supplements are available for figure 10:

**Figure supplement 1.** CotH extended phylogeny.

**Figure supplement 2.** Features of prokaryotic LGT.

expression. Despite some global similarities to animal regulation, we observed significant differences in how these regulatory layers are deployed in *Creolimax* and animals. For example, the population of lincRNAs that we described in *Creolimax* does not show a major enrichment near developmental genes nor a highly specific cell type-dependent expression, both hallmarks of metazoan lincRNAs (*Ulitsky and Bartel, 2013*; *Gaiti et al., 2015*). The differential processing of the intron-rich genes during alternative splicing in *Creolimax* is dominated by IR and not ES. Although this is similar to most other eukaryotic groups including sponges (*Fernandez-Valverde et al., 2015*), it contrasts with alternative splicing in more complex animals, which predominantly involves ES (*Sebé-Pedrós et al., 2013*; *Mcguire et al., 2008*). Therefore, while alternative splicing in *Creolimax* is likely to contribute to the regulation of gene expression, it does not provide a greater expansion of the proteome by generating multiple protein isoforms from individual genes as in animals (*Irimia and Roy, 2014*).

Intriguingly, while this complex genome regulation in *Creolimax* often involves similar gene toolkits to those employed in animal development, it is the unicellular amoeboid stage and not the multinucleate stage that is defined by multicellularity-related activities. This is the case for most differentially expressed TFs, signaling pathways, and adhesion molecules, which are characteristically associated with animal development and multicellularity. In fact, our results indicate that the transcriptional profile of the multinucleate stage of *Creolimax* is more similar to those of highly proliferative cell types in humans, despite the obvious differences in cell morphology and cell division strategy (cell divison vs. coenocytic nuclear division). Thus, we consider that the multinucleate stage of *Creolimax* can be regarded as a highly specialized proliferative cell type. This cell type can be considered to function in a manner analogous to stem cells, with the undifferentiated nuclei dividing in the multinucleate cell before differentiation into amoeboid cells closes the lifecycle. This would suggest that separation of functions as crucial as self-renewal and differentiation can occur in a unicellular context in a temporal manner, which pre-dates the exclusive ability of multicellular organisms to engulf both functionally distinct cell types within a single entity (*Arendt, 2008*; *Hemmrich et al., 2012*). Drastic differences in cell structure, morphology, function, and molecular signatures found between stages in protistan relatives of animals indicate that cell stages can be considered cell types according to established definition (57 and see author's response section for an in-depth discussion on this topic).

In addition to assessing the impact of genome regulation in *Creolimax*, our multiple genome-wide approaches reveal other novel aspects of ichthyosporean biology. For example, we show that *Creolimax* has undergone secondary adaptation to a specialized osmotrophic feeding mode, shaping its secretome and genome through both gene duplication and acquisition of bacterial genes by LGT. These observations highlight the uniqueness of the different holozoan lineages, each representing derived specializations from an ancestral state. However, these specializations of unicellular holozoans are achieved through a largely common genetic toolkit shared with animals, including several signaling pathways, TFs and adhesion molecules (*Richter and King, 2013*; *Suga et al., 2013*). Moreover, many of the components of this toolkit are assembled in co-regulated gene modules preserved since the common ancestor of all holozoans, suggesting that recurrent recruitment of full co-regulated gene programs underlies the evolution of lineage-specific cell types and developmental modes (*Newman, 2012*). Additional complementary insights on the evolution of developmental modes will be provided by studying the immediate out-group of the Holozoa and Holomycota (fungi + Discicristoidea) (*Torruella et al., 2015*). The Holomycota show a wide variety of developmental modes, ranging from aggregative fruiting body formation to several modes of coenocytic development (*Stajich et al., 2009*; *Brown et al., 2009*).

We have shown that the diversity of cellular behaviors and morphologies observed in holozoan lifecycles is likely to have evolved from lineage-specific specializations. A widely accepted hypothesis states that the origin of animals involved a clonally dividing organism, similar to choanoflagellate colonies, that subsequently evolved specialized cellular differentiation (*King, 2004*; *Nielsen, 2008*). A competing hypothesis, however, suggests that pre-existing cell types and their associated molecular mechanisms were integrated in the spatiotemporal developmental dynamics of the last common ancestor of all animals (*Mikhailov et al., 2009*). Our data suggests a third mixed model, in which the capacity to build differentiated cell types and transient multicellular entities was not a rare feature in pre-metazoan evolution. Nevertheless, the rise of animal multicellularity is not directly homologous to any ancestral developmental mode, and it may be seen as another derived specialization involving

the integration of ancestral molecular modules and their associated cell behaviors into one single multicellular entity. Consequently, our results reveal the importance of obtaining complementary data from multiple lineages before significant insights can be gained into the organism that took the first steps on the road towards complex animal multicellularity.

## Materials and methods

### Cell culture and nucleic acid extraction

*Creolimax fragrantissima* cells (available from the Canadian Centre for the Culture of Microorganisms under accession numbers CCCM 101 – 107) were grown axenically in liquid medium (marine broth Difco 2216) at 12°C. To obtain biological replicates for the RNA extraction, three independent cell lines were isolated from distinct colonies grown on a marine agar plate (marine agar Difco 2216). Those initial isolates were then grown in liquid medium (marine broth Difco 2216). After one pass, new 1/10 subcultures (10 ml) were initiated and grown statically for 5 days in 25 ml flasks. When the cells became confluent on the fifth day, they were scratched and passed into 50 ml flasks with an additional 25 ml of fresh medium. These 50 ml flasks were then grown for 48 hr with gentle agitation (150 rpm), allowing the mature coenocytes to form clumps. Then, the 50 ml flasks were filtered using a 5.0 µm Isopore membrane filter (Millipore ) and collected into a 50 ml Falcon tube . As only amoebas pass through the 5 µm filter step, filtered cells were then immediately pelleted by centrifugation at 1500 rcf for 3 min and harvested to get the RNA from the amoeboid stage. For the multinucleate-stage RNA, filtered cells were re-cultured in a new 50 ml flask, grown for 24 hr and harvested by centrifugation at 3000 rcf for 3 min. For all cell lines and stages, the RNA was extracted using Trizol reagent (Life Technologies, Carlsbad, CA) with a further step of DNAseI (Roche) to avoid gDNA contamination, and then purified using RNeasy columns (Qiagen).

### Genome sequencing, assembly and annotation

We generated 1.7 million 454 single reads and 34 million Illumina 5kb mate-pair reads (both after trimming, totaling 3.4G bp). Those were combined and preassembled with a Newbler 2.7 assembler (Roche). The mitochondrial DNA sequence was removed before the assembly. Using the pairing information of the Illumina mate-pair reads, the 846 pre-scaffolds were broken at unreliable positions found by REAPR 1 (*Hunt et al., 2013*) and re-assembled by SSPACE 2 (*Boetzer et al., 2011*). Some of the N-stretches within the scaffolds were filled by Gapfiller (*Boetzer and Pirovano, 2012*). This array of improvement tools assembled the pre-scaffolds into 196 sequence pieces. We then used the pairing information of the mate-pair reads for further assembly improvement, breaking and reconnecting the scaffolds. Finally, we obtained an assembly with 82 final scaffolds, of which 29 were short (<1000 bp) fragments.

To predict the protein-coding genes from the whole genome sequence, we used Augustus 2.7 (*Stanke et al., 2008*) combined with RNA-seq data (see details on this data in 'RNA-seq and differential expression analysis' section). We followed the protocol described here: http://bioinf.uni-greifswald.de/bioinf/wiki/pmwiki.php?n=IncorporatingRNAseq.Tophat. Briefly, we pooled all the RNA-seq samples and mapped them to the genome using Tophat2 (*Kim et al., 2013*), using the resulting introns to train Augustus *ab initio* predictions in an iterative process. The resulting predictions were manually screened in a genome browser and compared to the spliced-aligned reads resulting from Tophat2. We further validated our predicted annotation, comparing the data to a set of genes that we had previously cloned by RT-PCR and rapid amplification of cDNA ends PCR, including highly expressed genes (e.g. Histone 2B, Tubulin *beta*) and lowly expressed genes (e.g. Myc, Grainyhead, p53, Src Tyrosine Kinase).

Moreover, we used the mapped transcriptome to perform a genome-guided Trinity assembly (*Haas et al., 2013*). Those transcripts were then used to annotate the UTRs of the protein-coding genes resulting from the Augustus annotation step. The elongation of the UTRs was done using PASA (*Haas, 2003*). The transcripts that did not overlap with Augustus annotations were then searched against the NCBI non-redundant protein database using tBlastX. Those that retrieved significant hits (e-value <10e-3) and had clear open reading frames were then manually annotated as protein-coding transcripts. The resulting annotation retrieved 8695 genes, which are available here: 10.6084/m9.figshare.1403592.

To functionally annotate the genes, we used Blast2GO (*Gotz et al., 2008*), searching the protein sequences against the NCBI non-redundant database using a BLASTP threshold of 10e-6 and the default InterProScan settings. We also performed a PfamScan search with PFAM A database version 26 using the default gathering threshold parameters (*Punta et al., 2012*). As a result, 6814 genes were functionally annotated.

## RNA-seq and differential expression analysis

100-base paired-end libraries were constructed using the TruSeq Stranded mRNA Sample Prep kit (Illumina, San Diego, CA, USA). The libraries were sequenced in two lanes of an Illumina HiSeq2000 instrument at the CRG genomics unit (Barcelona, Spain). We obtained 417 million reads that were then mapped to the genome using Tophat2 (*Kim et al., 2013*), resulting in an average mapping of 82%. Raw gene counts and FPKM (fragments per kilobase of exon per million fragments mapped) values were obtained using Cufflinks2 (*Trapnell et al., 2013*). Differential expression analysis was performed by comparing the three replicates from each stage using DEseq (*Anders and Huber, 2010*) (threshold 5e-5), EdgeR (*Robinson et al., 2010*) (threshold 5e-5), Cuffdiff2 (*Trapnell et al., 2013*) (threshold 5e-5), and NOIseq (*Tarazona et al., 2011*) (threshold 0.8). Only the genes that were identified as differentially expressed with at least three methods were taken, resulting in 956 genes. Data can be downloaded from GEO GSE68616.

GO enrichments were obtained using the Topology-Weighted method in Ontologizer (*Bauer et al., 2008*) taking a *P* value lower than 0.01 as a threshold (see full list in *Figure 2—source data 1*). The resulting Gene Ontology (GO) enrichments were then visualized as a network in Cytoscape using the Enrichment map plug-in (*Merico et al., 2010*). Enrichment map plug-in connects GO terms according to gene annotation; therefore connected GOs belong to the same set of genes with multiple associated GOs. We used network connectivity between enriched GO terms as a criterion to collapse GO redundancy, shading general GO groups in more inclusive categories as seen in *Figure 2*. Inclusive categories complementary relied on the functional similarity between GOs based on GO definitions (e.g. distinct unconnected aminoacid metabolic pathways are collapsed in 'Amino-acid metabolism' inclusive category). Additionally, PFAM domain enrichments were calculated using a Fisher's exact test implemented in R, taking a threshold of 0.01. PFAM enrichments were also visualized as a network using the Enrichment map plug-in, where connected nodes reflect domain presence in the same genes.

## Genome-wide analysis of alternative splicing

Identification and quantification of ES (including events with single or multiple cassette exons and microexons of 3–15 nucleotides [nt]) and IR were performed as previously described (*Braunschweig et al., 2014*; *Irimia et al., 2014*). For ES, we used two complementary approaches. First, we implemented a 'splice site-based module', which utilizes the joining of all hypothetically possible exon–exon junction (EEJ) forward combinations from annotated and de novo splice sites (as described in [*Han et al., 2013*]). To identify splice sites de novo, for each annotated splice site donor/acceptor, we scanned two downstream/upstream introns for potential splice-site acceptors/donors that would constitute a novel EEJ. Next, we mapped our RNA-seq data to this library of all potential EEJs, and considered 'novel splice sites' those supported by at least five reads mapped to multiple positions of the EEJ. Then, we implemented our recently described 'microexon module' (*Irimia et al., 2014*), which also includes de novo searching of pairs of donor and acceptor splice sites in intronic sequences to detect novel, very short (3–15 nt) microexons. For IR, we used our recently described pipeline (*Braunschweig et al., 2014*), which employs a comprehensive set of reference sequences comprising exon–intron junctions (EIJs), intron sequences (if introns were longer than 200 nt, only a mid-intron segment of 200 nt was used), and EEJs formed by intron removal. Introns were classified as retained when there was a balanced accumulation of reads mapping to 5′ and 3′ EIJs and the intron body sequence, relative to the EEJ sequence. The level of retention was calculated based on PIR, which is the percentage of transcripts from a given gene in which the intron sequence is present.

In all modules, quantification of alternative sequence inclusion in the transcripts is derived only from junction reads (either EEJs or EIJs). To increase the fraction of mapping junction reads within each RNA-Seq sample, each read was first split into 50 nt read groups using a sliding window of 25

nt. Therefore, each 100 nt (replicates a and b) and 125 nt (replicate c) read produces 3 and 4 over-lapping reads, respectively. In addition, both read mates from the paired-end sequencing were pooled. These 50 nt split reads were then mapped to the genome using Bowtie (*Langmead et al., 2009*) with –m 1 –v 2 parameters (unique mapping with no more than two mismatches). Reads that mapped to the genome were discarded for ES quantifications. For quantification, only one random count per read group (i.e. all sub-reads coming from the same original read) was considered to avoid multiple counting of the same original sequenced molecule. In addition, for all modules and alternative splicing types, final read counts were corrected for the number mappable positions in each EEJ or EIJ following the formula:

$$Corrected\_EEJ_{count} = EEJ_{count} * \frac{Maximum_{mappability}}{EEJ_{mappability}}$$

where $EEJ_{count}$ is the number of read groups mapped to the EEJ, $Maximum_{mappability}$ the maximum number of mapping positions that any EEJ can have for reads of length 50 nt (i.e. 35 positions), and $EEJ_{mappability}$ the number of positions that can be mapped uniquely to the EEJ using specific bowtie parameters (–m 1 –v 2), and thus $EEJ_{mappability} \leq Maximum_{mappability}$ (see [*Barbosa-Morais et al., 2012*; *Han et al., 2013*] for details).

The different modules to detect and quantify AS have been integrated into vast-tools (https://github.com/vastgroup/vast-tools; species key "Cfr"). Associated files can be downloaded at http://vastdb.crg.eu/libs/vastdb.cfr.31.1.15.tar.gz.

We used a threshold of ≥20 PIR in at least one stage for positive intron retention events. As a threshold for ES events we used skipping rates below 90% (measured using the metric Percent Spliced In, PSI) in at least one stage. In both cases, the minimum coverage allowed was 20 reads per splice junction. To evaluate differential alternative splicing, we used differences over 15 PSI or PIR between stages, allowing a standard deviation of <10 between replicates.

## lincRNA annotation

From the genome-guided Trinity assembly (see *Genome sequencing, assembly and annotation*) we further analyzed the transcripts that did not retrieve any significant TBLASTX hit against the NCBI non-redundant database (e-value > e-3) and were more than 200 bp long. To avoid lineage-specific protein-coding genes, we performed an additional TBLASTX search against six frame translations of the de novo assembled transcriptomes of several closely related species (*S. arctica, Ichthyophonus hoferi, Pirum gemmata, Amoebidium parasiticum, Abeoforma whisleri, Corallochytrium limacisporum, C. owczarzaki, S. rosetta, Monosiga brevicollis*) and the protein coding genes of *Creolimax*, filtering out the positive hits (e-value < e-3). From the remaining transcripts, we performed a RfamScan_2 search against RFAM 11.0 (*Burge et al., 2013*) to annotate all known ncRNAs (e.g. U6, 18S, 28S). Additionally, we used the coding potential calculator (*Kong et al., 2007*) to discard all those transcripts with putative coding potential (coding potential score < −0.5). With the final list of transcripts, we selected those that did not overlap with gene+UTRs annotations or were close to uncertain assembly regions (multi-N stretches). For all those transcripts that were in a head-to-tail orientation regarding protein-coding genes, we manually inspected those that had an intergenic distance shorter than 1000 bp to the nearby gene to filter out misannotated UTRs. Additionally, we discarded transcripts overlapping repetitive regions of the genome. Finally, we collapsed all the remaining transcripts into single loci and quantified their expression level using Cuffcompare and Cufflinks (*Trapnell et al., 2013*). From the resulting 2661 loci, to avoid noisy transcription, we filtered out all those that did not have at least 5 FPKM in at least one sample, and over 1 FPKM in any other sample.

To detect putative homology, lincRNAs were searched using BLASTN against the same list of closely related species described above. Differential expression analysis of the lincRNAs was done using the same parameters as for the coding genes (see RNA-seq and differential expression analysis). Consequently, we only accepted lincRNA loci as differentially expressed when they were identified by at least 3 out of 4 methods. We validated 6 out of 6 lincRNA using RT-PCR (see below). Finally, we used the same pipeline to detetect alternative splicing events in coding genes, using a minimum coverage of 10 reads for each splice junction.

## Reverse transcription-polymerase chain reaction

To validate lincRNAs and alternative splicing events, RNA samples obtained as described in *Cell culture, gDNA and RNA extraction* were reverse transcribed to cDNA using SMARTer cDNA kit (CloneTech). For both stages, the same amount of initial purified RNA was used (1 µg). Pairs of primers with melting temperatures close to 60°C were designed to capture the lincRNA and the alternative splicing events, and the PCR was performed using Expand high-fidelity Taq polymerase (Roche). Validations of IR and ES events were preformed using primer pairs spanning the neighboring constitutive exons. Quantification of alternative sequence inclusion levels from gel band intensity was done using ImageJ software (*Schneider et al., 2012*).

## Comparative transcriptomic cross-species clustering

For the cross-species comparison, we first identified one-to-one orthologs in the proteomes of *Creolimax, Capsaspora*, and *Salpingoeca* using the Multiparanoid pipeline (*Alexeyenko et al., 2006*). We trimmed all RNA-seq datasets into the same length (50 bp) and only mapped the left reads when paired-end data was available. We then obtained cRPKM (corrected by mappability) values only for the subset of orthologs, transformed them to log2(cRPKM +1) and further normalized the expression data using quantile normalization. Hierarchical clustering ('complete' method) of samples was obtained by comparing pairwise distances based on Spearman correlation coefficients in R. To obtain the neighbor-joining trees and bootstrap supports across the samples, we used the 'ape' package in R (*Paradis et al., 2004*). For the *Creolimax*/human comparison we followed the same methodology, using the data detailed in *Figure 6—source data 1*. The PCA of the expression data was performed as implemented in R 'prcomp' function. GO enrichments were obtained as described in RNA-seq and differential expression analysis section.

## Secretome proteomics and in silico prediction

To obtain the secretome sample for proteomics, we cultured *Creolimax* cells in liquid medium (marine broth Difco 2216) at 12°C for 5 days and allowed the cells to attach to the bottom of the flask. The medium was then replaced with artificial seawater to avoid excessive protein contamination, and the culture was incubated for another 24 hr. Then, we collected the medium by gently tilting the flask to avoid collecting attached cells. The medium was immediately centrifuged at 10,000 rcf for 2 min, and the supernatant was collected and filtered twice through a 0.2 µm filter. The filtered medium was concentrated by ultrafiltration using a molecular weight cut-off membrane (Vivaspin 6, 3000 MW; Sartorius, Gottingen, Germany) and quantified using the BCA Protein Quantification Kit (Thermo Fisher Scientific, San Jose, CA). The resulting protein extract was digested with 5 µg of trypsin (cat # V5113, Promega ) (overnight, 37°C). Finally, 2 µg of the sample was analyzed using an LTQ-Orbitrap XL mass spectrometer (Thermo Fisher Scientific) coupled to an EasyLC (Thermo Fisher Scientific (Proxeon), Odense, Denmark) at the CRG proteomics unit (Barcelona, Spain). All data were acquired with Xcalibur software v2.2. Proteome Discoverer software suite (v1.4, Thermo Fisher Scientific) and the Mascot search engine (v2.5, Matrix Science (*Perkins et al., 1999*)) were used for peptide identification and quantification. The data were searched against a database containing *Creolimax* proteome, a list of common contaminants, and all the corresponding decoy entries. Resulting data files were filtered for false discovery rate (FDR)<0.05. Finally, we discarded the contaminants and all the proteins that were identified by less than two unique peptides, resulting in a list of 91 proteins.

To obtain the list of in silico secretome components, we performed an initial search step using SignalP 3.0 (*Dyrløv Bendtsen et al., 2004*) (D-cutoff = 0.450) to identify all the proteins with a canonical signal peptide. We then performed a search step with TMHMM v.2.0 (*Krogh et al., 2001*) to discard all those proteins with a transmembrane domain downstream of the first 60 amino acids. We also filtered out proteins tagged to the mitochondria using TargetP v.1.1 (*Emanuelsson et al., 2007*), and proteins with endoplasmic reticulum retention signal with c-terminal motifs KDEL or HDE [LF] and GPI-anchored proteins using PredGPI (*Pierleoni et al., 2008*). The resulting list comprised 453 proteins. PFAM and GO enrichment analyses were performed as described in *RNA-seq and differential expression analysis*.

## Ortholog identification, phylogeny and LGT detection

In order to obtain orthology assignments for the various gene families analyzed in this study, we used a phylogeny-based pipeline. First, we used PFAM domain information to obtain all the members of a gene family across a database comprising 108 eukaryotic proteomes. Then, proteins were aligned using MAFFT software with L-INS-i parameters (*Katoh and Standley, 2013*). The resulting alignments were automatically trimmed using trimAl v1.2 (*Capella-Gutierrez et al., 2009*) (-gt 0.7) and phylogenies were obtained using RAxML v8.0 (*Stamatakis, 2014*) (LG model, *gamma* distribution, 100 bootstrap supports) and Phylobayes 3 (*Lartillot et al., 2009*) (LG model, ran until two chains converged). Tree visualization and annotation was performed using iTol v2 (*Letunic and Bork, 2011*).

This pipeline was slightly modified to detect LGT cases. Instead of using PFAM, we gathered close orthologs of all the proteins in the genome by performing a BLASTP search against the NCBI non-redundant database plus the 108 eukaryotic genomes. Only those proteins that retrieved lower e-values for bacterial/archaeal hits were selected for downstream analysis. Those proteins were then separately searched using BLASTP against all bacteria in nr, all Archaea in nr, and the 108 eukaryotic proteomes. We selected those sequences that had at least 25 hits with an e-value under e-10, selecting a maximum of 50 proteins for bacteria, and 25 for archaea and eukaryotes. We got rid of redundancy using CD-HIT (*Li and Godzik, 2006*) by filtering for 0.95 identity and then performed alignment, trimming, and phylogeny as described in the general pipeline above. The resulting trees were analyzed manually, taking LGT positives when *Creolimax* (and other ichthyosporean) sequences branched within bacterial clades with nodal bootstrap supports over 70%. Finally, we manually checked that the resulting LGT genes were found in distinct parts of the genome and not in genomic singletons. For those LGT candidates without introns and not found in any other ichthyosporean (therefore, *Creolimax* specific) we further checked if they were located in scaffolds with other genes containing introns. To further discard bacterial contaminations, the neighboring genes to LGT candidates were blasted against NCBI nr to verify their eukaryotic origins. When the immediate neighbor did not retrieve any hit, the following gene was searched.

## Acknowledgements

We would like to thank the proteomics unit in the CRG for help in the analysis of the secretome, especially Eduard Sabidó and Guadalupe Iglesias, and the genomics unit of CRG for help in the RNA-seq library preparation and sequencing. We also thank Arnau Sebé-Pedrós, Xavi Grau-Bové and Ozren Bogdanovic for critical reading and discussion of the manuscript. Finally, we are grateful to Meritxell Antó for the technical support provided throughout the project.

## Additional information

### Funding

| Funder | Grant reference number | Author |
|---|---|---|
| European Research Council | ERC-2007-StG-206883 | Alex de Mendoza<br>Hiroshi Suga<br>Iñaki Ruiz-Trillo |
| Ministerio de Economía y Competitividad | BFU-2011-23434 | Alex de Mendoza |
| European Research Council | ERC-2012-Co -616960 | Alex de Mendoza<br>Hiroshi Suga<br>Iñaki Ruiz-Trillo |
| Japan Society for the Promotion of Science | Research Activity | Hiroshi Suga |
| Center for Genomic Regulation | Core funding | Jon Permanyer<br>Manuel Irimia |

The funders had no role in study design, data collection and interpretation, or the decision to submit the work for publication.

## Author contributions

AdeM, Conception and design, Acquisition of data, Analysis and interpretation of data, Drafting or revising the article; HS, MI, Acquisition of data, Analysis and interpretation of data, Drafting or revising the article; JP, Acquisition of data, Analysis and interpretation of data; IRT, Conception and design, Drafting or revising the article

## Author ORCIDs

Alex de Mendoza, http://orcid.org/0000-0002-6441-6529

# Additional files

## Major datasets

The following datasets were generated:

| Author(s) | Year | Dataset title | Dataset URL | Database, license, and accessibility information |
| --- | --- | --- | --- | --- |
| Alex de Mendoza, Iñaki Ruiz-Trillo | 2015 | Developmental transcriptomics of the ichthyosporean Creolimax fragrantissima | http://www.ncbi.nlm.nih.gov/geo/query/acc.cgi?acc=GSE68616 | Publicly available at NCBI Gene Expression Omnibus (Accession no: GSE68616). |
| Alex de Mendoza, Hiroshi Suga, Iñaki Ruiz-Trillo | 2015 | Creolimax fragrantissima genome data | http://dx.doi.org/10.6084/m9.figshare.1403592 | Publicly available at FigShare. |

The following previously published datasets were used:

| Author(s) | Year | Dataset title | Dataset URL | Database, license, and accessibility information |
| --- | --- | --- | --- | --- |
| Broad Institute | 2009 | Capsaspora owczarzaki ATCC 30864 | http://www.ncbi.nlm.nih.gov/bioproject/?term=PRJNA20341 | Publicly available at the NCBI BioProject database (Accession no: PRJNA20341). |
| Broad Institute | 2011 | Salpingoeca rosetta ATCC50818 transcriptome project | http://www.ncbi.nlm.nih.gov/bioproject/62005 | Publicly available at NCBI BioProject database (Accession no: 62005). |

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
