## [Decision Letter]

Thank you for submitting your work entitled "Complex transcriptional regulation and independent evolution of fungal traits in a close relative of animals" for peer review at *eLife*. Your submission has been favorably evaluated by Diethard Tautz (Senior Editor), Alejandro Sánchez Alvarado (Reviewing Editor), and three reviewers.

The reviewers have discussed the reviews with one another and the Reviewing Editor has drafted this decision to help you prepare a revised submission.

This manuscript by de Mendoza et al. presents a comprehensive analysis of the genome and stage-specific transcriptomes of a member of the earliest holozoan lineage: the ichthyosporean *Creolimax frangrantissima*. As such, this study puts forward intriguing inferences about the evolution of animal and fungal attributes. The extensive suite of analyses incorporated into this manuscript could have led the authors to simply produce a catalog of observations; however, by focusing the interpretation of the results on biological inferences, the manuscript strays minimally. An example is provided by the range of analyses presented ranging from differences in expression levels, lincRNAs, and alternative splicing between the amoeboid and multinucleate stages, to detection of coregulation of various biological pathways, to analyzing the secretome.

Perhaps the most interesting findings pertain to the analysis of the integrin adhesome, filopodial, and postsynaptic machinery. There's been much written about the presence/absence of the components of these cellular structures and the authors present a creative analysis that gives meaningful insight about the functions of these pathways in holozoan lineages that are not easily amenable to functional studies.

In sum, from the analyses reported by de Mendoza et al., two main themes emerge: (i) *Creolimax* possesses some genomic and transcriptomic (co-expression) features that suggest regulatory features associated with metazoan multicellularity have a more ancient origin; and (ii) that *Creolimax* displays a range of lineage-specific innovations that separate it from other opisthokonts. From these, it is inferred that although each major lineage of holozoan has evolved a unique manner of being "muliti-celled", their last common ancestor possessed the genomic foundation to allow for a rudimentary multi-phased life cycle with a colonial stage.

Essential revisions:

1) The authors seek to understand the regulatory differences between the amoeboid and multinucleate stages in *C. frangrantissima* and how they relate to cell type specification in metazoans. It appears that the multinucleate stage shows higher expression of genes for cell proliferation, which are in turn enriched for intron retention in the amoeboid stage, confirming that the multinucleate stage has highly proliferative nuclei (since it was already known that these cells are diving cells). The authors should provide more information on how the GO categories and PFAM domains were assigned to larger groupings (grey circles in Figure 2 and Figure 2—figure supplement 1). The authors rest their conclusions on these categories, so this should be done in a more objective manner, e.g. the LIM domain does not have to have an actin cytoskeleton function.

2) The authors should be careful not to conflate stages with cell types. In metazoan biology, different cell types have different biological functions that go beyond the capacity to divide or not (e.g., intestinal stem cells and hematopoetic stem cells have different epigenomic landscapes that facilitate the different functions of these cells, yet both cells are able to divide and though both cells will show differential expression of genes when they are quiescent vs. proliferating, those phases are considered to be just different stages of the same cell type). The statement in the second paragraph of the Discussion is not supported by these data – *C. frangrantissima* cells clearly go through a proliferative stage, but these functions are not "segregated" because the amoeboid and proliferative cells don't exist in one entity, they are separated temporally.

3) The analyses throughout the manuscript are very clear in showing the different biological functionalities of various stages in C*. fragrantissima* and *C. owczarzaki*, specifically, in a PCA of expression levels of orthologous genes in the two species, PC2 and PC3 separate cells based on proliferation and motility/adhesion. The authors' focus on thinking of these stages as "types" leads to some overstatements and reconsidering these statements in view of the biological functions would be more meaningful, e.g., the title of the section "Evidence of species-specific cell type evolution in Holozoa". Also the statement at the end of this subsection is stronger than needed – don't we know from the yeast cell cycle that genes for human cell proliferation are conserved eukaryotic genes?

4) Throughout the article, the authors should clearly state what we already knew before this study. e.g., *C. fragrantissima* multinucleate cells may use similar machinery to human proliferating cells, but we know that the cell cycle is a highly conserved phenomenon, well-studied in yeast (the outgroup).

5) The authors use word the "fungal" referring to features of *Creolimax* which are similar to these of fungi (in the title, Abstract and in the beginning of the subsection “The secretome of *C. fragrantissima* shows convergent adaptations to a specialized osmotrophic lifestyle”). It makes me a bit uneasy. I would suggest using terms "fungal-like" or perhaps simply "osmotrophic".

6) In the second paragraph of the Introduction the syncytial stage of *Creolimax* is compared to D*rosophila's* syncytial blastoderm and to glass sponge tissues. I feel the similarity to *Drosophila* is a much better one, as in glass sponges the syncytial organization is the final step of development, rather than transient stage.

In the subsections “Recurrent prokaryotic lateral gene transfer events shaped the *C. fragrantissima* secretome” and”Ortholog identification, phylogeny and LGT detection”, the authors address the issue of possible bacterial contaminations presenting a lateral gene transfer. Perhaps the most convincing way would be to check whether the genes which are likely results of LGT are present on the same genomic scaffolds as unquestionably eukaryotic genes?

7) In the subsection “Genome sequencing, assembly and annotation” the authors describe a combination of Augustus and RNA-Seq data to generate protein predictions. This section is rather brief, and could be made more valuable to the readers if we could learn some more details of the efficiency/correctness of their pipeline. For example, given that PASA can easily miss a gene if its transcripts do not align perfectly on the genome, how big was the fraction of Trinity transcripts which could be aligned to the genome but were rejected by PASA? How were the Augustus models verified not only during the training phase, but also the final gene models? It would be recommended to at least manually compare a sub-set of Augustus models with transcripts from Trinity, or computationally compare of e.g. intron-exon boundaries and UTRs predicted by Augustus with those in Trinity dataset.

[Editors' note: further revisions were requested prior to acceptance, as described below.]

Thank you for resubmitting your work entitled "Complex transcriptional regulation and independent evolution of fungal traits in a close relative of animals" for further consideration at *eLife*. Your revised article has been favorably evaluated by Diethard Tautz (Senior Editor), a Reviewing Editor, and three reviewers. The manuscript has been improved but there are some remaining issues that need to be addressed before acceptance, as outlined below.

Two reviewers have still some comments that need to be addressed before the final version of the manuscript can be accepted. Please look particularly carefully into the comments concerning the GO/PFAM analysis. When resubmitting the revised version, please add a detailed reply to the remaining comments.

*Reviewer #1:*In this revised version of the paper, de Mendoza et al. have addressed most of my previous comments, but two points remain:

I think the GO/PFAM analysis needs some more clarification or trimming. The authors have now provided a table with p-values and GO categories that the differentially expressed genes were assigned to. This is very helpful, but I still am not sure how the authors came up with the "more inclusive functions". For example, GO:0060026 (name: convergent extension) does not have "actin cytoskeleton" as a parent category in the GO database. Of the thousands of human genes that fall in this GO:0060026, some are *sfrp* and *wnt*, which by no means have molecular functions directly related to actin cytoskeleton. So how was this GO term placed under the grey cloud of "actin cytoskeleton"? Are these classifications based on subjective decisions made by the authors? If so, I think that this should be made clear in the Methods section. Furthermore, I think this undercuts this point of this analysis – GO assignments and PFAM annotations provide unbiased ways of looking at large genomic datasets. If the authors then make subjective decisions on the inclusive categories, then they may be introducing bias. The authors also gave the example of the gene of the integrin adhesome that did not obtain the correct GO terms, which the authors corrected manually. However, they could do this for a set of genes they are very familiar with, but what about the many other genes that they are not as familiar with? This manual curation is certainly valuable when studying a specific set of genes, but, in analyzing large datasets, where an unbiased method is supposed to give you biological insight, this seems arbitrary.

The authors have presented good reasons to think of the multinucleate and amoeba stages as different cell types in their rebuttal. However, this nuanced discussion is missing from the manuscript itself. I think the authors should add a very explicit description of these two cell types with their differing features before they start talking about the different "cell types". Also, though they have edited the second paragraph of the Discussion, I think a more in depth discussion of cell-types vs. cell-stages, that explores the ideas in the authors’ rebuttal, would greatly enhance the manuscript.

*Reviewer #3:* For the most part, I am happy with the changes made in this revision.

However the general statement that ichthyosporeans are close relatives of animals, which is stated a number of times in the manuscript, is still misleading. If the metazoans and their closest unicellular relatives, the choanoflagellates, are estimated to have diverged 150-200 million years before the emergence of the crown Metazoa some 7-800 Mya and the ichthyosporeans diverged well before this (before but probably closer to the metazoan-fungal divergence time), it seems stating they are close relatives of animals is very misleading.

It would benefit the reader if the Opisthokont relationships can be more clearly articulated, and if the authors can provide accurate estimates of divergence times of the lineages shown in Figure 1. When discussing the various 'developmental' modes of holozoans it would be appropriate to include some discussion of fungi, as they are the outgroup to the Holozoa. This, in turn, should lead to a more balanced discussion about the evolution of development and multicellularity in the Holozoa.

---

## [Author Response]

*Essential revisions: 1) The authors seek to understand the regulatory differences between the amoeboid and multinucleate stages in C. frangrantissima and how they relate to cell type specification in metazoans. It appears that the multinucleate stage shows higher expression of genes for cell proliferation, which are in turn enriched for intron retention in the amoeboid stage, confirming that the multinucleate stage has highly proliferative nuclei (since it was already known that these cells are diving cells). The authors should provide more information on how the GO categories and PFAM domains were assigned to larger groupings (grey circles in Figure 2 and Figure 2—figure supplement 1). The authors rest their conclusions on these categories, so this should be done in a more objective manner, e.g. the LIM domain does not have to have an actin cytoskeleton function.*

The groupings were done using the network connectivity between different nodes as an indicator of redundancy using the EnrichmentMap plugin (Merico et al., 2010, PLoS ONE) in Cytoscape. In that software the edges between the nodes are the genes shared by the enriched GOs, therefore clusters of connected GOs belong to a similar set of genes. From those clusters of GOs we obtained a more inclusive function by assessing which genes and functions were more redundant. For example, in the case of Protein Kinase activity or GTPase function, many GOs were associated to those more inclusive functions. Some unconnected nodes were grouped in the inclusive categories by similarity of the function. As a case example, Focal Adhesion GO appears unconnected to the rest of GOs related to the Integrin pathway. This is because one gene of the Integrin Adhesome did not retrieve the integrin GOs in the Blast2GO search, but it had a Focal Adhesion GO in the annotation step. Both GOs should share the same genes. It should be noted that Blast2GO performance in distant non-model organisms such as *Creolimax* has some limits of detection if compared to its performance in a species with close relatives sequenced and annotated in the databases (e.g. vertebrates or insects). Thus, we preferred to group the resulting GOs in inclusive categories rather than curating the GOs for each gene manually. To address the reviewers’ concerns, we now include the GO categories depicted in Figure 2 in [Supplementary-material SD1-data], with the respective p-values and group classification.

In the case of the domains, we used the information available in each of those PFAM domain webpage descriptions to curate inclusive categories that summarize the data in meaningful sets. In the cases where a domain has unspecific functions (e.g. LIM or Ank_2), we used three criteria to assign it to an inclusive category: 1) we checked if the unspecific PFAM domain was found in the same gene with other specific domains, 2) we searched for the GOs of the differentially expressed genes containing the unspecific domain (e.g. GOs of LIM-containing genes enriched in the amoeboid stage) 3) we used a network connectivity criteria, checking redundancy of genes between GO and PFAM categories in a mixed Enrichment map network. This “guilty-by-association” procedure produced the final categories, but as reviewers suggest we now exclude LIM from “Actin cytoskeleton”. Despite LIM domain was associated with “Cell migration associated with gastrulation” GO in the combined network, only 2 of 9 genes had that GO associated. We now also explain these criteria in Figure 2 and Figure 2—figure supplement 1 figure legends.

*2) The authors should be careful not to conflate stages with cell types. In metazoan biology, different cell types have different biological functions that go beyond the capacity to divide or not (e.g., intestinal stem cells and hematopoetic stem cells have different epigenomic landscapes that facilitate the different functions of these cells, yet both cells are able to divide and though both cells will show differential expression of genes when they are quiescent vs. proliferating, those phases are considered to be just different stages of the same cell type). The statement in the second paragraph of the Discussion is not supported by these data –* C. frangrantissima *cells clearly go through a proliferative stage, but these functions are not "segregated" because the amoeboid and proliferative cells don't exist in one entity, they are separated temporally.*

This is, of course an interesting open conceptual debate. In our opinion, the cell type concept can potentially be used within a unicellular context, and specifically to *C. fragrantissima* life cycle. Both under the reviewer’s definition and under the definition by Detlev Arendt in his 2008 Nature Review Genetics: “What is a cell type and how can cell types be compared? By definition, any cell type has special physiological or structural characteristics. The aim of comparative study of cell type characteristics is to elucidate the evolutionary diversification of cell types (cell typogenesis) by detecting the similarities and differences between them. Physiological and structural characteristics will be reflected by cell type-specific gene expression at some point, and therefore comparisons can be based on differential expression profiling data”. In this case we observe two completely different cell types, not only restricted to their ability to proliferate. The multinucleate stage is not only a coenocyte (many nuclei sharing a single cytoplasm) but also has a thick cell wall reminiscent of those of fungi and has unique ultra-structural features (see Marshall et al., 2008, Protist). Moreover it has different biological functions to the amoeboid stage, not only division but also encystation, as multinucleate cells are able to stop growing under non-favourable circumstances. On the other hand, the amoeba stage lacks a cell wall, it is mono-nucleated and has a dispersal function.

Another line of evidence towards the segregation of cell types in this species is that in some rare cases, a mature multinucleate stage can bypass the amoeboid stage, just entering into the multinucleate stage again (as reported in Marshall et al., 2008, Protist and Suga and Ruiz-Trillo, 2013, Developmental Biology). Moreover, the closely related species from the Sphaeroforma genus, do not have this amoeboid stage (see Marshall and Berbee, 2013, Protist). Thus, both the molecular and morphological differences are huge; enough – in our opinion – to merit the designation of different cell types separated temporally. We suggest keeping the denomination of “cell type”, but clarifying the ideas further. For example, we have rephrased the second paragraph of the Discussion which now reads as: “This would suggest that separation of functions as crucial as self-renewal and differentiation can occur in a unicellular context in a temporal manner, which pre-dates the exclusive ability of multicellular organisms to engulf both functionally distinct cell types within a single entity”.

*3) The analyses throughout the manuscript are very clear in showing the different biological functionalities of various stages in* C. fragrantissima *and* C. owczarzaki*, specifically, in a PCA of expression levels of orthologous genes in the two species, PC2 and PC3 separate cells based on proliferation and motility/adhesion. The authors' focus on thinking of these stages as "types" leads to some overstatements and reconsidering these statements in view of the biological functions would be more meaningful, e.g., the title of the section "Evidence of species-specific cell type evolution in Holozoa". Also the statement at the end of this subsection is stronger than needed – don't we know from the yeast cell cycle that genes for human cell proliferation are conserved eukaryotic genes?*

Following the reasoning above, we think the cell type nomenclature could be used in this context, although we have toned down the text significantly. Specifically, a choanoflagellate cell, a filopodiated amoeba, a cystic coenoctye and a crawling amoeba are completely distinct in function, physiology and, obviously, gene expression profiles. In an evolutionary perspective, it is conceivable that the ancestors of some of those species might have had more cell types than those currently displayed. In fact, some extant species might have some cell types still unknown because they are not seen in culture conditions. However, if the definition of cell type is restricted to cells co-existing in one multicellular entity with some defined cell lineages, then we are open to include the changes. Our point is to broaden the definition of cell type, as we think that the frontiers between multicellular and unicellular species are blurrier than commonly thought.

We now have rephrased the end of the subsection “Evidence of species-specific cell type evolution in Holozoa” according to reviewers’ suggestion. Now it reads “These results suggest that a signal from the evolutionarily conserved machinery for cell proliferation in eukaryotes can be detected from direct expression pattern comparisons between the multinucleate stage of *C. frangrantissima* and highly proliferative human cell types.”

Accordingly, we have also rephrased the sentence in the Discussion: “In fact, our results indicate that the transcriptional profile of the multinucleate stage of *C. fragrantissima* is more similar to those of highly proliferative cell types in humans, despite the obvious differences in cell morphology and cell division strategy (cell divison vs. coenocytic nuclear division).”

*4) Throughout the article, the authors should clearly state what we already knew before this study. e.g.,* C. fragrantissima *multinucleate cells may use similar machinery to human proliferating cells, but we know that the cell cycle is a highly conserved phenomenon, well-studied in yeast (the outgroup).*

We agree. We think that the previously modified sentences now include awareness about the conservation of some general features of the cell cycle throughout eukaryotes. In that same sentence we have included two citations to recent reviews on the topic: “Cell cycle control across the eukaryotic kingdom” by Harashima et al. (2013, Trends in Cell Biology) and “Evolution of networks and sequences in eukaryotic cell cycle control” by Cross et al. (2011, Philos Trans R Soc Lond B Biol Sci.) However, we would like to note that recent data (covered in both those reviews) suggest that the cell cycle described in yeast (and in fact, in most fungi) is only analogous to the cell cycle of animals and not homologous. In fact, gene content analysis on Cyclins and CDKs suggest that unicellular holozoans might be better out-groups to understand the ancestral cell cycle of animals than yeast in terms of gene orthology (see Suga et al., 2012, Nature Communications). However, discussing this issue is out of the scope of this paper.

*5) The authors use word the "fungal" referring to features of* Creolimax *which are similar to these of fungi (in the title, Abstract and in the beginning of the subsection “The secretome of* C. fragrantissima *shows convergent adaptations to a specialized osmotrophic lifestyle”). It makes me a bit uneasy. I would suggest using terms "fungal-like" or perhaps simply "osmotrophic".*

We now use “fungal-like” as suggested.

*6) In the second paragraph of the Introduction the syncytial stage of* Creolimax *is compared to* Drosophila's *syncytial blastoderm and to glass sponge tissues. I feel the similarity to* Drosophila *is a much better one, as in glass sponges the syncytial organization is the final step of development, rather than transient stage.*

*In the subsections “Recurrent prokaryotic lateral gene transfer events shaped the* C. fragrantissima *secretome “and”Ortholog identification, phylogeny and LGT detection”, the authors address the issue of possible bacterial contaminations presenting a lateral gene transfer. Perhaps the most convincing way would be to check whether the genes which are likely results of LGT are present on the same genomic scaffolds as unquestionably eukaryotic genes?*

We have now deleted the reference to the glass sponge tissues. Regarding the LGT genes, we have many sources of evidence to rule out putative bacterial contamination. First of all, *Creolimax* grows in axenic conditions. Therefore, we are pretty sure about the purity of our gDNA. Moreover, in most cases we are detecting the LGT events not only in *Creolimax*, but also in other ichthyosporean species (135 of the genes, in front of 35 only observed in *Creolimax,* as shown in Figure 10—figure supplement 2). The other ichthyosporeans sequences have been generated in the same axenic conditions, and have been sequenced in different platforms in different institutes (Broad Institute, BGI and CRG). Therefore those sequences are pretty unlikely to come from contamination from other material during library construction. In the case of the 35 LGTs only detected in *Creolimax*, 17 have at least one intron supported by RNA-seq data, discarding bacterial origin. The other 18 intron-less genes are distributed in 17 scaffolds, that range from 29 to 843 genes and all those scaffolds contain genes with introns (in fact, genes belonging to genomic singletons were discarded for the analysis). Following the reviewers’ suggestion, we now have checked the top blast hit against the non-redundant database of NCBI of the neighbouring genes. In all the cases, genes with eukaryotic best hits were found surrounding the LGT candidates. We now have included this extra criterions exposed here in the “Materials and methods – Ortholog identification, phylogeny and LGT detection” subsection.

*7) In the subsection “Genome sequencing, assembly and annotation” the authors describe a combination of Augustus and RNA-Seq data to generate protein predictions. This section is rather brief, and could be made more valuable to the readers if we could learn some more details of the efficiency/correctness of their pipeline. For example, given that PASA can easily miss a gene if its transcripts do not align perfectly on the genome, how big was the fraction of Trinity transcripts which could be aligned to the genome but were rejected by PASA? How were the Augustus models verified not only during the training phase, but also the final gene models? It would be recommended to at least manually compare a sub-set of Augustus models with transcripts from Trinity, or computationally compare of e.g. intron-exon boundaries and UTRs predicted by Augustus with those in Trinity dataset.*

Yes, we are aware that PASA might miss some genes, but as we used a “genome-guided” Trinity assembly, the resulting transcripts mapped perfectly to the genome (citing Trinity’s manual, “reads are first aligned to the genome, partitioned according to locus, followed by de novo transcriptome assembly at each locus.”). Anyhow, we used Trinity+PASA only to annotate the UTRs, feeding PASA with the annotation of CDS from Augustus and elongating the UTRs. Moreover, the information from UTRs is only used for the analysis of lincRNA, to avoid identifying as a lincRNA a miss-annotated UTR.

In the case of Augustus models for the Coding Sequence (we did not use it to predict the UTRs), we went through several rounds of manually checking the genes in the IGV genome browser and comparing different parameters outputs with the spliced-alignments resulting from Tophat2 read mapping. Not only that, but we further used as a quality measure the recovery of a manually curated set of genes that we have amplified by RACE-PCR or RT-PCR from previous experiments, which covered both highly expressed genes such as Histone2B or Tubulin-*β* and lowly expressed genes such as Transcription Factors, including Grainyhead, Myc, NF-κ B or P53, or Tyrosine Kinases as Src. In fact, the gene models presented here are currently used in the lab to clone more genes of interest, usually retrieving positive results (e.g. Integrin Adhesome components).

To clarify this points for future readers, we have now extended the “Materials and methods – Genome sequencing, assembly and annotation” subsection in the manuscript.

[Editors' note: further revisions were requested prior to acceptance, as described below.]

Reviewer #1:

*In this revised version of the paper, de Mendoza et al. have addressed most of my previous comments, but two points remain: I think the GO/PFAM analysis needs some more clarification or trimming. The authors have now provided a table with p-values and GO categories that the differentially expressed genes were assigned to. This is very helpful, but I still am not sure how the authors came up with the "more inclusive functions". For example, GO:0060026 (name: convergent extension) does not have "actin cytoskeleton" as a parent category in the GO database. Of the thousands of human genes that fall in this GO:0060026, some are* sfrp *and* wnt*, which by no means have molecular functions directly related to actin cytoskeleton. So how was this GO term placed under the grey cloud of "actin cytoskeleton"? Are these classifications based on subjective decisions made by the authors? If so, I think that this should be made clear in the Methods section. Furthermore, I think this undercuts this point of this analysis – GO assignments and PFAM annotations provide unbiased ways of looking at large genomic datasets. If the authors then make subjective decisions on the inclusive categories, then they may be introducing bias. The authors also gave the example of the gene of the integrin adhesome that did not obtain the correct GO terms, which the authors corrected manually. However, they could do this for a set of genes they are very familiar with, but what about the many other genes that they are not as familiar with? This manual curation is certainly valuable when studying a specific set of genes, but, in analyzing large datasets, where an unbiased method is supposed to give you biological insight, this seems arbitrary.*

We agree this is an important point and we have tried to further clarify it in the text. As the reviewer states, it is very crucial for genome-wide studies that GO/PFAM analyses are done in an unbiased and objective manner. We have done so in our work, relying only on Blast2GO gene-to-term associations (Götz et al., 2008, Nucleic Acids Research) and Enrichment map GO term connections (Merico et al., 2010, Plos ONE). We believe the confusion may come from the fact that many genes have multiple associated GO terms. For example, in the case mentioned by the reviewer, the genes that have “convergent extension” as an associated GO also have “Actin binding” and “Rho GTPase binding” as associated categories in Creolimax Blast2GO annotation. Therefore, the signal from all those enriched GOs comes from the same set of genes, and this is why those GOs are connected in the network and shaded in an inclusive category. We would like to restate that we have not forced these connections – they are built by the “Enrichment map” plug-in for Cytoscape, which connects GOs through common genes. These GO networks simply allow visualization of GO redundancy, which is helpful when showing a very long list of GOs, as in this case (89 enriched categories). In other cases where the list of GOs is shorter we have shown all the significant categories in a bar graph plot (see Figure 3, Figure 5, Figure 6, Figure 8). Finally, in the case of Figure 2, we provide the full list of enriched GOs (as well as all enriched PFAM in Figure 2—figure supplement 1) so that the reader can take a look at our raw results. We have thus simply made an effort to show the results in the most informative manner. In that sense, we would like to clarify that none of the extended list of enriched GO categories contradicts our interpretation of the data.

We would also like to explain the case of the integrin adhesome cited by the reviewer. We did not manually correct the annotation of any gene. In fact, we wanted to specifically avoid this, as mentioned above, as we agree with the reviewer that unbiased methods are preferable in large dataset analysis. The GO focal adhesion and the group of GOs integrin-mediated signaling pathway/integrin complex/cell-matrix adhesion/lamellipodium/positive regulation of podosome assembly/cleavage furrow formation appear unconnected in the network and that is how we show it in Figure 2. But consulting the GO term definition and previous literature, all these categories belong to the same set of genes involved in Cell-ECM adhesion. We strongly believe that this kind of inclusion is not an “arbitrary” and “biased” decision, just reflects research and logical decisions made in order to summarize the data and present it in a meaningful way to the reader.

We have now provided more explicit and detailed explanations of these points in the corresponding Methods sections. Moreover, in the Results section we refer to the raw list of GOs (in the subsection “Transcriptional dynamics reveal differences between *Creolimax* multinucleate stage and animal development”) to draw attention to the reader of the availability of this data:

“Gene Ontology enrichments were obtained using the Topology-Weighted method in Ontologizer (Bauer et al., 2008) taking a P value lower than 0.01 as a threshold (see full list in [Supplementary-material SD1-data]). […] Additionally, PFAM domain enrichments were calculated using a Fisher’s exact test implemented in R, taking a threshold of 0.01. PFAM enrichments were also visualized as a network using the Enrichment map plug-in, where connected nodes reflect domain presence in the same genes.”

*The authors have presented good reasons to think of the multinucleate and amoeba stages as different cell types in their rebuttal. However, this nuanced discussion is missing from the manuscript itself. I think the authors should add a very explicit description of these two cell types with their differing features before they start talking about the different "cell types". Also, though they have edited the second paragraph of the Discussion, I think a more in depth discussion of cell-types vs. cell-stages, that explores the ideas in the authors’ rebuttal, would greatly enhance the manuscript.*

Now we include explicit sentences about cell stages in *Creolimax* in the Introduction (with referrals to Figure 1). It reads: *“*These stages have distinct physiological and structural characteristics, the amoeboid stage is mono-nucleated, non-diving and motile (Figure 1), while the multinucleate stage has a cell wall, a big central vacuole and does not move (Figure 1).” We also have expanded this argument in the Discussion section: “Drastic differences in cell structure, morphology, function and molecular signatures found between stages in protistan relatives of animals indicate that cell stages can be considered cell types according to established definitions (Fernandez-Valverde, Calcino and Degnan, 2015 and see Author’s response section for an in depth discussion on this topic).”

Reviewer #3:

*For the most part, I am happy with the changes made in this revision. However the general statement that ichthyosporeans are close relatives of animals, which is stated a number of times in the manuscript, is still misleading. If the metazoans and their closest unicellular relatives, the choanoflagellates, are estimated to have diverged 150-200 million years before the emergence of the crown Metazoa some 7-800 Mya and the ichthyosporeans diverged well before this (before but probably closer to the metazoan-fungal divergence time), it seems stating they are close relatives of animals is very misleading.*

The level of relatedness to the animal lineage is debatable. If we take the millions of years of divergence as a measure of relatedness, Ichthyosporeans and animals seem distantly related. But if we look at the whole array of known cellular life or eukaryotic diversity (more than 100 different lineages – see del Campo et al. Trends Ecol Evol, 29(5), 252–259), it is clear that only filastereans and choanoflagellates appear closer to the animal group than ichthyosporeans. Therefore, we think that “close relative” of animals is a valid term under the macro-evolutionary perspective used in this manuscript. Moreover, the major genomic similarities found between all the extant holozoan groups and stark differences with their sister group, the Holomycota, points towards this “close” relatedness from a genomic level, which is the main topic of this manuscript (how lineages with similar genomic complements achieve different forms and developmental modes). Nevertheless, we now accommodate the reviewer’s suggestions and we have deleted “close” from all the sentences where “close relative” appeared, including the title of the manuscript. As an exception, in the Abstract we now refer to the “the three closest animal relatives (ichthyosporeans, filastereans and choanoflagellates)”, which truly reflects the phylogenetic position of this groups, as there is no other know group branching closer to animals than those three.

*It would benefit the reader if the Opisthokont relationships can be more clearly articulated, and if the authors can provide accurate estimates of divergence times of the lineages shown in Figure 1. When discussing the various 'developmental' modes of holozoans it would be appropriate to include some discussion of fungi, as they are the outgroup to the Holozoa. This, in turn, should lead to a more balanced discussion about the evolution of development and multicellularity in the Holozoa*.

We are a bit confused about the first part of this reviewer's concern as the opisthokont relationships are clearly depicted in Figure 1, where there are no ambiguities about the placement of Fungi, Ichthyosporeans, Filastereans and Choanoflagellates in relation to animals. This phylogeny is depicted according to the latest and more comprehensive phylogenetic analysis designed to unravel Opisthokont relationships (Shalchian-Tabrizi et al., 2008, Plos ONE; Torruella et al., 2012, Molecular Biology and Evolution and Torruella et al., 2015, Current Biology).

Regarding the “accurate dating” of the divergence times, we agree that it is an important topic and there is plenty of literature on molecular clocks and eukaryotic phylogeny, but there is currently no consensus divergence time for the groups treated in this manuscript, or even for the origin of animals (published divergence times range from 698 Mya to 1200 Mya, nicely reviewed in Sharpe SC, Eme L, Brown MW and Roger AJ in the book chapter “Timing the Origins of Multicellular Eukaryotes Through Phylogenomics and Relaxed Molecular Clock Analyses”). Depending on the method, the parameters, the evolutionary models and the backbone of the phylogeny used to calculate those estimates, results keep changing and giving very disparate ages. Given that this manuscript’s focus does not include molecular clocks and there is not a consensus and “accurate” divergence time for the splitting of every holozoan group, we think including this information is detrimental to the manuscript, as it would attract criticism and controversy for a topic out of its focus. We have nonetheless mentioned this point in the figure legend and provide a reference to the review article mentioned above. It reads “Divergence times of the lineages shown in this figure range between 700 Mya (considered the latest estimates of animal origins) and 1200 Mya (earliest estimates of Opisthokont origins) (Li and Godzik, 2006)”.

Regarding discussing the fungal developmental modes, we would first like to highlight that not only the Fungi are the outgroup of the Holozoa, but all the Holomycota. Within the Holomycota, we find lots of different developmental mechanisms, from aggregative fruiting body formation in *Fonticula alba*, coenocytic development seen in some Chytrids to a huge variety of truly multicellular and multinucleate forms found in the fungal kingdom. We now mention this in the manuscript: “Further complementary insights on the evolution of developmental modes will be provided by studying the immediate out-group of the Holozoa, the Holomycota (Fungi + Discicristoidea) (Torruella et al., 2015). The Holomycota show a wide variety of developmental modes, ranging from aggregative fruiting body formation to several modes of coenocytic development (Stajich et al., 2009; Brown, Spiegel and Silberman, 2009).”